# Kinesin-6 Klp9 orchestrates spindle elongation by regulating microtubule sliding and growth

Lara Katharina Krüger[1]*, Matthieu Gélin[2], Liang Ji[1], Carlos Kikuti[1], Anne Houdusse[1], Manuel Théry[2,3], Laurent Blanchoin[2,3], Phong T Tran[1,4]*

[1]Institut Curie, PSL Research University, Sorbonne Université CNRS, UMR 144, Paris, France; [2]Institut de Recherche Saint Louis,U976 Human Immunology Pathophysiology Immunotherapy (HIPI), CytoMorpho Lab, University of Paris, INSERM, CEA, Paris, France; [3]Interdisciplinary Research Institute of Grenoble, Laboratoire de Physiologie Cellulaire & Végétale, CytoMorpho Lab, University of Grenoble-Alpes, CEA, CNRS, INRA, Grenoble, Paris, France; [4]Department of Cell and Developmental Biology, University of Pennsylvania, Philadelphia, United States

**Abstract** Mitotic spindle function depends on the precise regulation of microtubule dynamics and microtubule sliding. Throughout mitosis, both processes have to be orchestrated to establish and maintain spindle stability. We show that during anaphase B spindle elongation in *Schizosaccharomyces pombe*, the sliding motor Klp9 (kinesin-6) also promotes microtubule growth in vivo. In vitro, Klp9 can enhance and dampen microtubule growth, depending on the tubulin concentration. This indicates that the motor is able to promote and block tubulin subunit incorporation into the microtubule lattice in order to set a well-defined microtubule growth velocity. Moreover, Klp9 recruitment to spindle microtubules is dependent on its dephosphorylation mediated by XMAP215/Dis1, a microtubule polymerase, creating a link between the regulation of spindle length and spindle elongation velocity. Collectively, we unravel the mechanism of anaphase B, from Klp9 recruitment to the motors dual-function in regulating microtubule sliding and microtubule growth, allowing an inherent coordination of both processes.

*For correspondence:
lara-katharina.kruger@curie.fr (LKK);
phong.tran@curie.fr (PTT)

**Competing interests:** The authors declare that no competing interests exist.

## Introduction

Mitotic spindle assembly and function requires microtubules to undergo alternating periods of growth and shrinkage, together with the action of molecular motors, that can crosslink and slide apart spindle microtubules. A fine balance of microtubule dynamics and microtubule sliding is essential in all mitotic processes to achieve faithful chromosome separation (*Cande and McDonald, 1986*; *Cande and McDonald, 1985*; *Cheerambathur et al., 2007*; *Yukawa et al., 2019b*). Yet, while work has mainly been focused on each process independently, it is unclear how they are coordinated.

During anaphase B, the spindle elongates to push apart the spindle poles and separate the two chromosome sets (*Mallavarapu et al., 1999*; *Oegema et al., 2001*; *Roostalu et al., 2010*; *Scholey et al., 2016*; *Straight et al., 1998*; *Vukušić et al., 2017*). While in the fungus *Ustilago maydis* and Ptk1 cells, spindle elongation is mainly achieved through cortical pulling forces acting on astral microtubules (*Aist et al., 1993*; *Fink et al., 2006*; *Grill et al., 2001*), these forces are not present or dispensable in a plethora of other species. In yeast, *Caenorhabditis elegans*, *Drosophila melanogaster*, plants, and human, spindle elongation is realized by the generation of microtubule sliding forces at the spindle midzone (*Euteneuer et al., 1982*; *Khodjakov et al., 2004*; *Kiyomitsu and Cheeseman, 2013*; *Redemann et al., 2017*; *Sharp et al., 1999*; *Tolić-Nørrelykke et al., 2004*; *Vukušić et al., 2019*; *Yu et al., 2019*). The spindle midzone refers to the microtubule overlap at the

spindle center, formed by antiparallel-oriented microtubules (interpolar microtubules), which originate from the two opposite spindle poles (*Ding et al., 1993*; *Euteneuer et al., 1982*; *Mastronarde et al., 1993*; *McDonald et al., 1977*; *McIntosh and Landis, 1971*; *Ward et al., 2014*; *Winey et al., 1995*).

Microtubule sliding is promoted by members of the kinesin-5 family in most organisms (*Avunie-Masala et al., 2011*; *Brust-Mascher et al., 2009*; *Saunders et al., 1995*; *Sharp et al., 2000*; *Straight et al., 1998*). The tetrameric kinesin crosslinks antiparallel microtubules and slides them apart (*Avunie-Masala et al., 2011*; *Brust-Mascher et al., 2009*; *Kapitein et al., 2005*; *Saunders et al., 1995*; *Shimamoto et al., 2015*; *Straight et al., 1998*). By walking toward the microtubule plus-ends of each of the two microtubules, that it crosslinks, kinesin-5 can move the microtubules relative to each other (*Kapitein et al., 2005*). In fission yeast, this task is mainly performed by the kinesin-6 Klp9, which localizes to the spindle midzone upon anaphase B onset (*Fu et al., 2009*; *Krüger et al., 2019*; *Rincon et al., 2017*; *Yukawa et al., 2019a*). The kinesin-5 Cut7 also generates microtubule sliding forces, but to a lower extent than Klp9 (*Rincon et al., 2017*). In the absence of both sliding motors, spindle elongation is abolished, leading to 'cut' of chromosomes by the cytokinetic ring and mis-segregated chromosomes (*Rincon et al., 2017*).

Concomitantly with microtubule sliding, the interpolar microtubules have to grow at a rate that allows the spindle to elongate while maintaining the spindle midzone (*Cande and McDonald, 1986*; *Cande and McDonald, 1985*; *Cheerambathur et al., 2007*; *Masuda, 1995*; *Masuda and Cande, 1987*; *Saxton and McIntosh, 1987*). Microtubule growth has to occur with at least the velocity the spindle microtubules slide apart to keep the microtubule overlap, necessary for spindle stability. Microtubule dynamics during anaphase B have been shown to be regulated by CLASP, which promotes microtubule polymerization or increases the frequency of microtubule rescues (*Bratman and Chang, 2007*; *Maton et al., 2015*; *Pereira et al., 2006*). Moreover, silencing of the transforming acid coiled-coil (TACC) protein TACC3, which stabilizes microtubules together with XMAP215 (*Gergely et al., 2003*; *Kinoshita et al., 2005*; *Peset et al., 2005*), reduces the rate of spindle elongation by destabilizing midzone microtubules (*Lioutas and Vernos, 2013*). This strongly suggests that precise regulation of microtubule stability is necessary for proper spindle elongation. Accordingly, in animal cells, the kinesin-4 member Kif4a terminates anaphase B spindle elongation by promoting the inhibition of microtubule polymerization and thus restricting spindle midzone length (*Hu et al., 2011*). However, how the required velocity of microtubule growth is set precisely and how microtubule dynamics are coordinated with microtubule sliding to allow seamless spindle elongation remains enigmatic.

A straightforward way of coupling microtubule growth and sliding could involve a motor that sets equal sliding and polymerization speeds. Supporting this possibility, *Xenopus* kinesin-5 promotes microtubule polymerization in vitro (*Chen and Hancock, 2015*). The motor has been proposed to induce a curved-to-straight conformational transition of tubulin at microtubule plus-ends, thus promoting microtubule growth by virtue of a stabilizing effect (*Chen et al., 2019*). This may allow the motor to directly regulate the dynamic behavior of the microtubule tracks which it simultaneously slides. However, the biological significance of the polymerization function of kinesin-5 still has to be investigated in vivo.

Like kinesin-5 in most organisms, the kinesin-6 Klp9 promotes anaphase B spindle elongation in fission yeast (*Fu et al., 2009*; *Rincon et al., 2017*; *Yukawa et al., 2017*). Both motors form homotetramers and both slide apart microtubules. Moreover, Klp9 sets the velocity of spindle elongation in a dose-dependent manner (*Krüger et al., 2019*). Spindles elongate faster as the amount of motors increases at the spindle midzone. Thus, Klp9 is able to set the speed of spindle elongation. We, therefore, wondered if this is solely an effect of the microtubule sliding function or if Klp9 simultaneously regulates microtubule dynamics.

In the present study, we provide insight into the mechanism of anaphase B spindle elongation, from the regulation of Klp9 localization to the anaphase spindle, to the Klp9-mediated coordination of microtubule sliding and growth. Klp9 recruitment to spindle microtubules is regulated in a dephosphorylation-dependent manner by the microtubule polymerase Dis1, homolog of XMAP215 and ch-TOG (*Matsuo et al., 2016*). Furthermore, we provide evidence that, at the spindle, Klp9 performs two functions: the generation of microtubule sliding forces, as shown previously, and the regulation of microtubule dynamics. By using monopolar spindles in vivo and in vitro reconstitution assays with recombinant Klp9, we could show that the kinesin-6 is a crucial regulator of microtubule

growth. In fact, Klp9 increases or decreases the microtubule growth speed depending on the condition. Whereas, at low tubulin concentration, where microtubule growth is comparatively slow, Klp9 increases the microtubule growth speed, at high tubulin concentration, where microtubule growth is fast, Klp9 decreases microtubule growth speed in a dose-dependent manner. This suggests an unconventional mechanism by which Klp9 can promote and block tubulin dimer addition to the microtubule lattice. Eventually, Klp9 is able to set a well-defined microtubule growth velocity. With the dual-function of Klp9 in regulating the microtubule sliding and growth velocity, both processes are inherently coordinated to allow flawless spindle elongation.

## Results

### Monopolar spindles to study microtubule dynamics in anaphase B

We found monopolar spindles to be a useful tool to examine microtubule dynamics not only during early mitotic phases (*Costa et al., 2013*), but also during anaphase B. Monopolar spindles can be generated by the temperature-sensitive mutant *cut7-24* (*Hagan and Yanagida, 1992*; *Hagan and Yanagida, 1990*). Since the kinesin-5 Cut7 is essential to establish spindle bipolarity during spindle assembly by separating the two spindle poles (*Hagan and Yanagida, 1992*), inactivation of Cut7 at its restrictive temperature results in the formation of monopolar spindles (*Figure 1A*).

Live-cell imaging of fission yeast cells expressing α-tubulin (Atb2) linked to mCherry to visualize microtubules and Sid4 linked to GFP to mark spindle pole bodies (SPBs) showed the typical three phases of spindle dynamics in wild-type cells: prophase, metaphase and anaphase A, and anaphase B, during which the spindle dramatically elongates until it disassembles (*Figure 1A*; *Nabeshima et al., 1998*). In contrast, the *cut7-24* monopolar spindle underwent only two distinguishable phases (*Figure 1A*). First, short and rather dynamic microtubules emanated from the two unseparated spindle poles for approximately 2 hr. Following this phase, one or up to three long microtubule bundles grew from the spindle poles until they disassembled, just before the cell divided (*Figure 1A*, *Figure 1—figure supplement 1*). This extensive growth of microtubule bundles was reminiscent of the microtubule polymerization that is necessary to allow spindle elongation during anaphase B in bipolar spindles. The bundles grew up to an average length of 6.1 ± 1.0 µm, which is slightly longer than half of the average wild-type spindle length (11.6 ± 1.3 µm) (*Figure 1B*) with an average speed of 0.7 ± 0.2 µm/min, slightly faster than half of the velocity at which the bipolar wild-type spindle elongated (1.2 ± 0.2 µm/min) (*Figure 1B*, *Figure 1—source data 1*), while cell size is slightly decreased in the *cut7-24* mutant (*Figure 1—figure supplement 2*, *Figure 1—figure supplement 2—source data 1*). Thus, monopolar spindles appear to behave like half-spindles. Moreover, the growth velocity of individual bundles increased with final bundle length (*Figure 1C*, *Figure 1—source data 1*), a correlation we have previously observed for bipolar anaphase B spindles: longer spindles elongate with respectively higher velocities (*Krüger et al., 2019*). Together, the second phase of *cut7-24* spindle dynamics resembled anaphase B with the bundle growth velocity matching the speed of spindle elongation in bipolar spindles, and its final length being similar to half the final length of a bipolar anaphase spindle.

To further test whether monopolar spindles indeed proceed to anaphase B, we used the cyclin B Cdc13 linked to GFP as a marker. Cdc13 disappears from spindle microtubules just before anaphase B onset (*Decottignies et al., 2001*). Accordingly, we observed fading of the signal on wild-type spindle microtubules and *cut7-24* monopolar spindle microtubules just before the spindle or bundle was elongated (*Figure 1D*).

Next, we investigated the localization of crucial anaphase B spindle components in monopolar spindles. In bipolar spindles, the sliding motor Klp9 (kinesin-6) localizes to the spindle midzone from anaphase B onset (*Fu et al., 2009*; *Figure 1E*, left panel). In the *cut7-24* monopolar spindle, Klp9-GFP localized to the tip of the microtubule bundle once it started to elongate (*Figure 1E*, right panel, asterisk). The Klp9-GFP intensity profile obtained at three different time points of anphase B revealed that Klp9-GFP preferentially localized to the tip of the bundle from the first time point of bundle elongation (*Figure 1F*) and the intensity increased with anaphase B progression (*Figure 1F, G*, *Figure 1—source data 1*). Hence, while the bundle was growing Klp9 accumulated at its tip. In general, we observed a strong correlation of the Klp9-GFP intensity at the bundle tip and the bundle length (*Figure 1H*, *Figure 1—source data 1*).

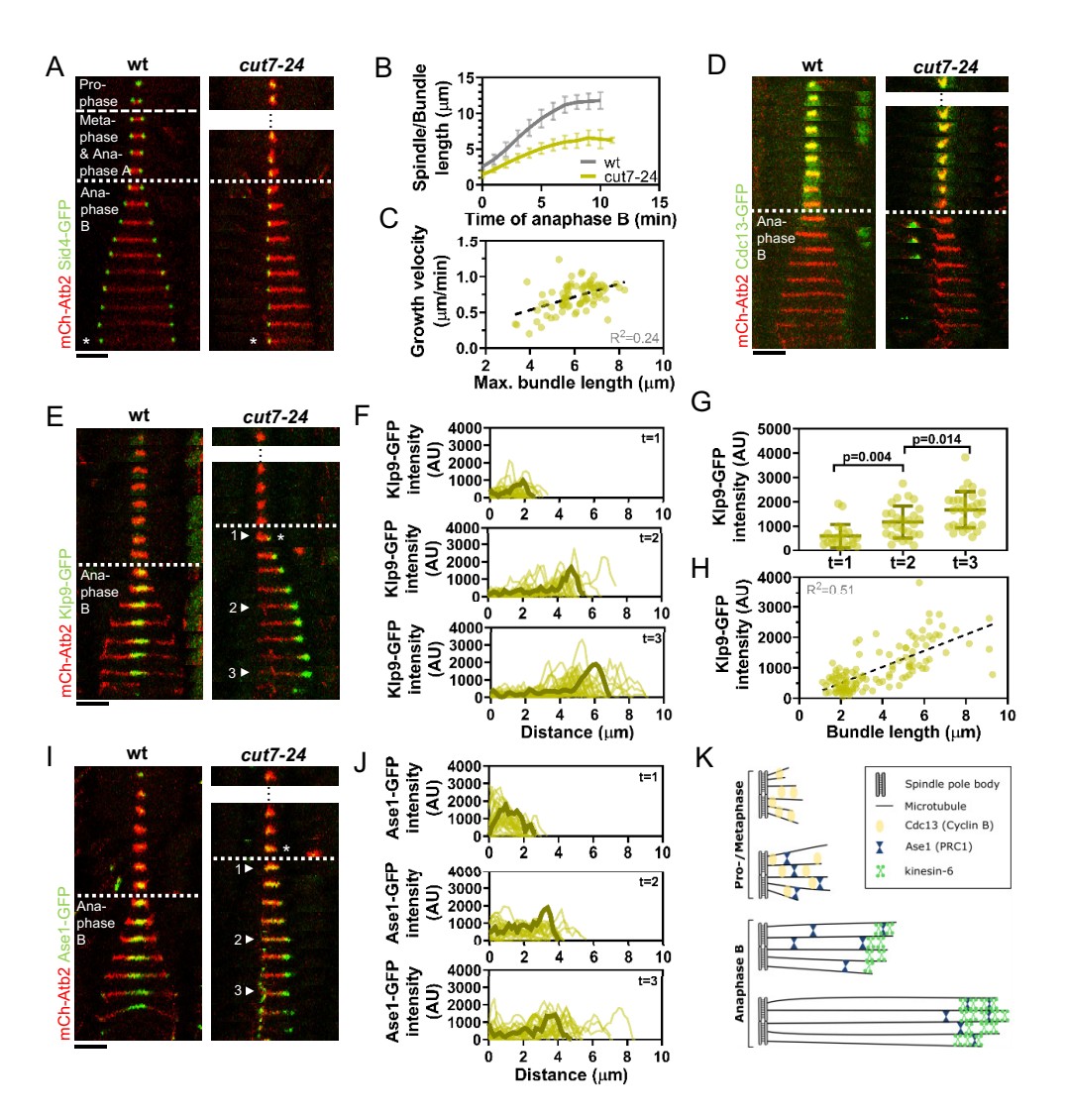

**Figure 1.** *Cut7-24* monopolar spindle as a tool to study microtubule dynamics during anaphase B. (A) Time-lapse images of wild-type and *cut7-24* cells expressing mCherry-Atb2 (tubulin) and Sid4-GFP (SPBs) at 37°C. The dashed line denotes the transition from prophase to metaphase. Dotted line denotes the transition to anaphase B. Asterisk marks spindle disassembly. (B) Comparative plot of anaphase B spindle dynamics of wild-type (n = 40) and bundle dynamics of *cut7-24* cells (n = 40) at 37°C. Bold curves correspond to the mean and error bars to the standard deviation. (C) Microtubule bundle growth velocity in *cut7-24* cells plotted against final bundle length. n = 79 microtubule bundles (D) Time-lapse images of wild-type and *cut7-24* cells expressing mCherry-Atb2 (tubulin) and Cdc13-GFP (cyclin B) at 37°C. (E) Time-lapse images of wild-type and *cut7-24* cells expressing mCherry-Atb2 (tubulin) and Klp9-GFP (kinesin-6) at 37°C. Asterisk marks the appearance of the Klp9-GFP signal on the microtubule bundle. Arrowheads 1, 2, and 3 correspond to the time points used for linescan analyses at t = 1 (first time point of anaphase B), t = 2 (5 min after anaphase onset), and t = 3 (last time point of anaphase B). (F) Intensity spectra obtained by linescan analysis of Klp9-GFP signals along the microtubule bundles of *cut7-24* cells at time points 1, 2, and 3. 0 µm on the x-axis marks the origin of the microtubule bundle at spindle pole bodies. Dark green lines display an exemplary spectrum. (G) Dot plot comparison of the Klp9-GFP intensity on microtubule bundles of *cut7-24* cells at time points 1 (n = 22), 2 (n = 28), and 3 (n = 30). Dark green lines display the mean and standard deviation. p-values were calculated using Mann–Whitney U test. (H) Klp9-GFP intensities along microtubule bundles of *cut7-24* cells plotted against bundle length. n = 122 (I) Time-lapse images of wild-type and *cut7-24* cells expressing mCherry-Atb2 (tubulin) and Ase1-GFP at 37°C. Asterisk marks the appearance of the Ase1-GFP signal on the microtubule bundle. Arrowheads 1, 2, and 3 correspond to the time points used for linescan analysis at t = 1 (first time point of anaphase B), t = 2 (5 min after anaphase onset), and t = 3 (last time point of anaphase B). Scale bar, 5 µm. (J) Intensity spectra obtained by linescan analysis of Ase1-GFP signals along microtubule bundles at time points 1, 2, and 3. 0 µm on the x-axis marks the origin of the microtubule bundle at spindle pole bodies. Dark green lines display an exemplary spectrum. (K) Model of *cut7-24* monopolar spindles displaying phase I, termed pro-metaphase, and phase II corresponding to anaphase B as judged by the absence of the Cdc13-GFP signal and the presence of Klp9-GFP and Ase1-GFP in the long microtubule bundles. In (A, D, E) and (I), each frame corresponds

*Figure 1 continued on next page*

*Figure 1 continued*

to 1 min interval. Dotted lines denote the transition to anaphase B. Scale bar, 5 μm. In (C) and (H), data was fitted by linear regression (dashed line), showing the regression coefficient ($R^2$) and the slope m. Data from n cells was collected from at least three independent experiments.

The online version of this article includes the following source data and figure supplement(s) for figure 1:

**Source data 1.** Numerical data used for *Figure 1B, C, F, G, H, and J*.
**Figure supplement 1.** *Cut7-24* cells expressing mCherry-Atb2 (tubulin) and Sid4-GFP (SPBs) at 37°C.
**Figure supplement 2.** Dot plot comparison of cell length (μm) of wild-type and *cut7-24* cells at mitosis onset.
**Figure supplement 2—source data 1.** Numerical data used for *Figure 1—figure supplement 2*.

Last, we analyzed the localization of the conserved microtubule bundler Ase1/PRC1. Ase1 cross-links the antiparallel-overlapping microtubules at the spindle center and stabilizes the spindle structure (*Janson et al., 2007*; *Loïodice et al., 2005*; *Yamashita et al., 2005*). In the wild-type spindle, Ase1-GFP localized to the spindle midzone just before anaphase B onset and onward, as previously reported (*Loïodice et al., 2005*; *Figure 1I*, left panel). In the monopolar spindle, Ase1-GFP similarly localized to the spindle just before the microtubule bundle started to grow (*Figure 1I*, right panel, asterisk). The signal was spread all along the bundle at early time points of bundle elongation (*Figure 1I,J*; timepoint 1), and accumulated at the bundle tip at later time points (*Figure 1I,J*; timepoint 2 and 3, *Figure 1—source data 1*). This could result from Ase1 binding all along the parallel microtubule lattice initially, then being carried to the bundle tip by a plus-end directed motor. Klp9 may transport Ase1, as the motor has previously been shown to physically interact with the cross-linker (*Fu et al., 2009*).

Taken together, besides unseparated spindle poles, *cut7-24* monopolar spindles proceed to anaphase B, as judged by the absence of the Cdc13-GFP signal, and the presence of Klp9-GFP and Ase1-GFP on growing microtubule bundles (*Figure 1K*). Initiation of anaphase B, despite unsegregated DNA in this mutant (*Hagan and Yanagida, 1990*), appears to be the result of a leaky checkpoint at the metaphase-to-anaphase transition. After 1–2 hr, *cut7-24* cells will proceed to anaphase B irrespectively of unseparated spindle poles or DNA. During this phase long microtubule bundles are polymerized with a bundle growth velocity that equals half of the spindle elongation velocity of bipolar spindles, suggesting that microtubule dynamics are not altered in the mutant and regulated independently of microtubule sliding forces. Therefore, monopolar spindles constitute a suitable tool to study microtubule dynamics during anaphase B.

## The kinesin-6 Klp9 affects microtubule growth during anaphase B

Using monopolar spindles, we could now analyze the effect of Klp9 on microtubule dynamics. As reported earlier (*Fu et al., 2009*), deletion of *klp9* strongly decreased the speed of bipolar spindle elongation in anaphase B (*Figure 2A*). Deletion of *klp9* in the *cut7-24* background prevented the formation of long microtubule bundles (*Figure 2B*). While anaphase B microtubule bundles reached a maximum length of 6.1 ± 1.0 μm in the presence of Klp9, the bundles only grew up to 2.7 ± 0.5 μm upon *klp9* deletion (*Figure 2C*, *Figure 2—source data 1*). Moreover, the growth velocity of the bundles was strongly reduced in *klp9* deleted cells (*cut7-24*: 0.7 ± 0.2 μm/min; *cut7-24 klp9Δ*: 0.1 ± 0.1 μm/min) (*Figure 2C*, *Figure 2—source data 1*). To test whether this effect is dose dependent, we used a shut-off strain, in which the expression of *klp9* is strongly reduced, but some Klp9 molecules are still present. Indeed, the phenotype was slightly less dramatic as compared to the deletion of *klp9* (*Figure 1C*). The bundles reached a maximum length of 3.2 ± 0.8 μm and grew with an average velocity of 0.2 ± 0.2 μm/min. Together, this suggests that Klp9 is involved in the regulation of microtubule growth during anaphase B.

However, Klp9 has been previously implicated in the regulation of the metaphase-to-anaphase B transition (*Meadows et al., 2017*). We thus had to rule out that the observed reduction of microtubule growth is not a consequence of an impaired anaphase B transition in monopolar spindles upon *klp9* deletion. To do so, we analyzed the localization of Cdc13-GFP. In a bipolar *klp9Δ* spindle, Cdc13-GFP disappeared from the spindle just before initiation of anaphase B spindle elongation (*Figure 2D*). In the *cut7-24 klp9Δ* mutant, Cdc13-GFP also disappeared from the spindle microtubules, but still no long microtubule bundle was formed (*Figure 2D*). Hence, the observed phenotype can be attributed to the Klp9 function in microtubule growth regulation during anaphase B.

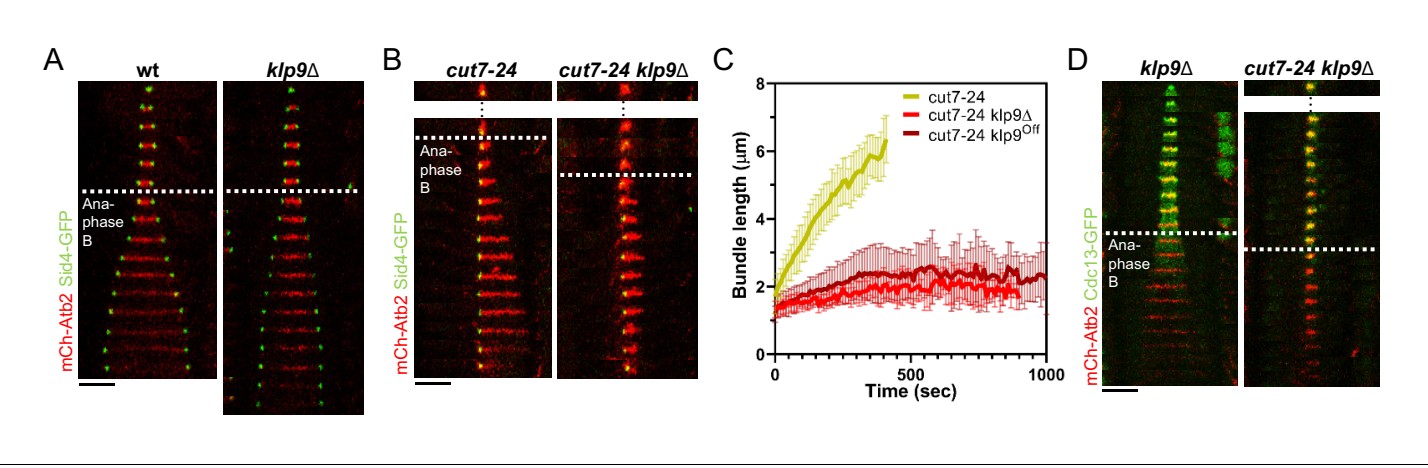

**Figure 2.** Klp9 promotes microtubule growth during anaphase B in monopolar spindles. (**A**) Time-lapse images of wild-type and *klp9Δ* cells expressing mCherry-Atb2 (tubulin) and Sid4-GFP (SPBs) at 37˚C. (**B**) Time-lapse images of *cut7-24* and *cut7-24 klp9Δ* cells expressing mCherry-Atb2 (tubulin) and Sid4-GFP (SPBs) at 37˚C. (**C**) Comparative plot of microtubule bundle dynamics in *cut7-24* (n = 40), *cut7-24 klp9Δ* (n = 40), and *cut7-24 klp9*^Off (n = 40) at 37˚C. Bold curves correspond to the mean and error bars to the standard deviation. (**D**) Time-lapse images of *klp9Δ* and *cut7-24 klp9Δ* cells expressing mCherry-Atb2 (tubulin) and Cdc13-GFP (cyclin B) at 37˚C. In (**A–B**) and (**D**), each frame corresponds to 1 min interval. Dotted lines denote the transition to anaphase B. Scale bar, 5 µm. Data from n cells was collected from at least three independent experiments.

The online version of this article includes the following source data for figure 2:

**Source data 1.** Numerical data used for *Figure 2C*.

## The XMAP215 family member Dis1 displays a similar effect on microtubule bundle growth as Klp9

Klp9 could either regulate microtubule dynamics by itself or indirectly through another microtubule-associated protein (MAP). To probe the involvement of other proteins, we analyzed the impact of the deletion of several candidates on microtubule bundle growth. Namely, the conserved microtubule bundler Ase1, which has been proposed to recruit Klp9 and other spindle components, such as CLASP to the mitotic spindle (*Bratman and Chang, 2007*; *Fu et al., 2009*); the EB1 homolog Mal3, which impacts microtubule dynamics in vitro (*Bieling et al., 2007*; *des Georges et al., 2008*; *Katsuki et al., 2009*; *Matsuo et al., 2016*); the two members of the XMAP215 family, Dis1 and Alp14, which act as microtubule polymerases (*Al-Bassam et al., 2012*; *Matsuo et al., 2016*); and the TACC protein Alp7, which interacts with Alp14 and enhances its polymerase activity (*Hussmann et al., 2016*; *Sato et al., 2004*).

The deletion of *ase1*, *mal3*, *alp14*, and *alp7* still allowed the formation of comparatively long microtubule bundles in the *cut7-24* background (*Figure 3A,B*). In all cases, the growth velocity of microtubule bundles was significantly reduced (*Figure 3B,C*, *Figure 3—source data 1*). However, only the deletion of *dis1* led to a decrease of the bundle growth velocity comparable to the deletion or shut-off condition of *klp9* (*Figure 3B,C*, *Figure 3—source data 1*). Moreover, maximum bundle length was reduced upon deletion of *ase1*, *dis1*, and *alp7*, with the strongest decrease observed upon *dis1* deletion (*Figure 3D*, *Figure 3—source data 1*).

We note that the deletion of *alp14* and *alp7* often resulted in the restoration of spindle bipolarity. This is in agreement with the model, that a balance between microtubule dynamics and the action of Cut7 is crucial for the establishment of spindle bipolarity (*Yukawa et al., 2019b*). Here, we analyzed the fraction of cells in which spindle bipolarity could not be restored, and spindles remained monopolar.

Taken together, of the tested candidates, the deletion of *dis1* gave rise to a similar phenotype as the deletion or shut-off condition of *klp9*. Similar to the deletion of *klp9*, upon *dis1* deletion the transition to anaphase B is not impaired, as indicated by the disappearance of the Cdc13-GFP signal (*Figure 3—figure supplement 1*). Thus, like Klp9, Dis1 seems to be involved in the regulation of microtubule growth during anaphase B and may act in the same pathway with Klp9.

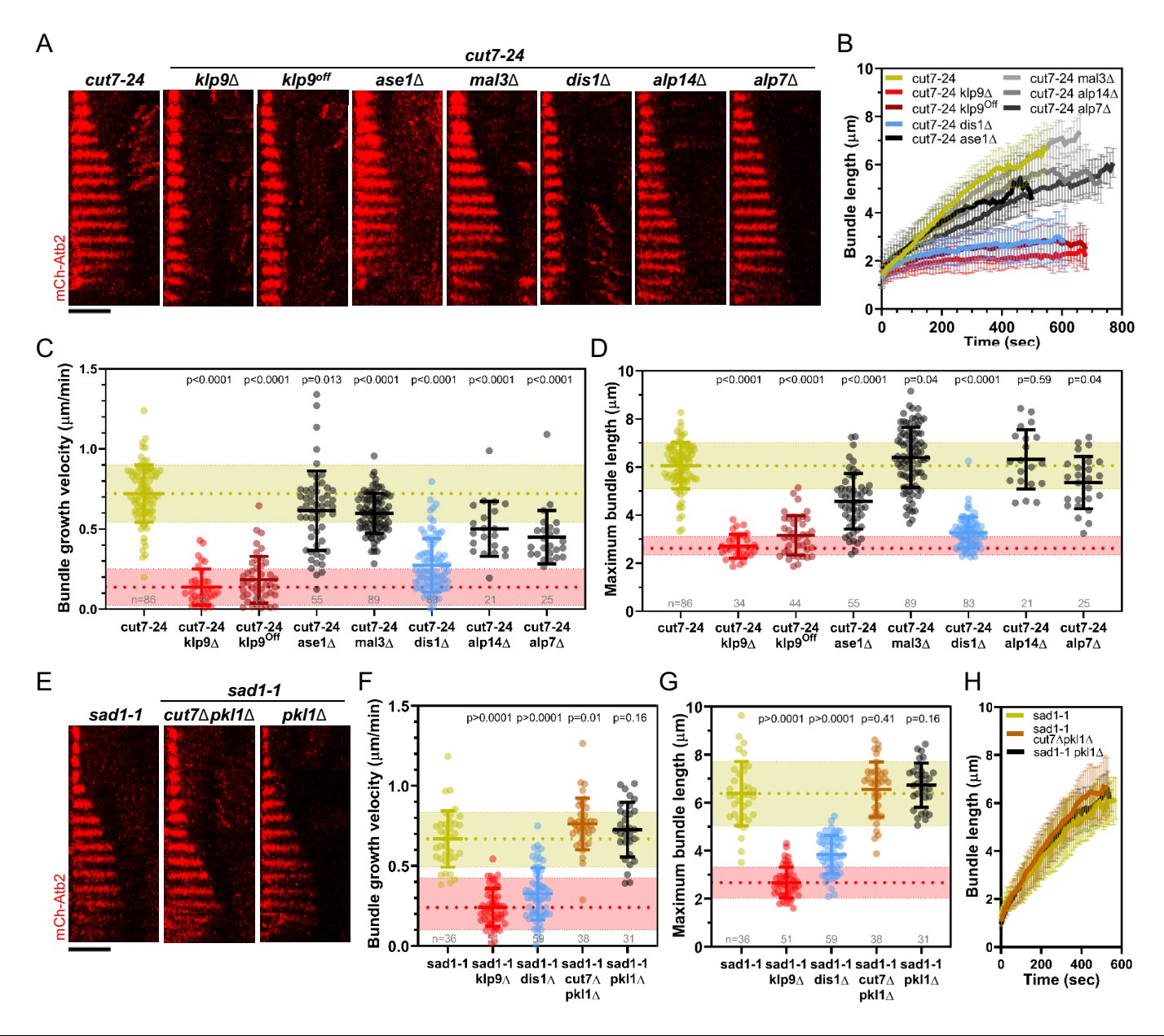

**Figure 3.** Deletion of *dis1* decreases microtubule bundle growth velocity and bundle length during anaphase B in monopolar spindles. (**A**) Time-lapse images of *cut7-24*, *cut7-24 klp9Δ*, *cut7-24 klp9^Off^, klp9^Off^ ase1Δ*, *cut7-24 mal3Δ*, *cut7-24 dis1Δ*, *cut7-24 alp14Δ*, and *cut7-24 alp7Δ* cells expressing mCherry-Atb2 (tubulin) at 37°C. (**B**) Comparative plot of microtubule bundle dynamics in *cut7-24* (n = 40), *cut7-24 klp9Δ* (n = 40), *cut7-24 klp9^Off^* (n = 40), *cut7-24 ase1Δ* (n = 39), *cut7-24 mal3Δ* (n = 40), *cut7-24 dis1Δ* (n = 39), *cut7-24 alp14Δ* (n = 20), and *cut7-24 alp7Δ* cells (n = 25) at 37°C. (**C**) Dot plot comparison of microtubule bundle growth velocity during anaphase B. (**D**) Dot plot comparison of maximum microtubule bundle length during anaphase B. (**E**) Time-lapse images of *sad1-1*, *sad1-1 cut7Δpkl1Δ* and *sad1-1 pkl1Δ* cells expressing mCherry-Atb2 (tubulin) at 37°C. (**F**) Dot plot comparison of microtubule bundle growth velocity during anaphase B. (**G**) Dot plot comparison of maximum microtubule bundle length during anaphase B. (**H**) Comparative plot of microtubule bundle dynamics in *sad1-1* (n = 39), sad1-1 *cut7Δpkl1Δ* (n = 38), and *sad1-1 pkl1Δ* (n = 31) at 37°C. In (**A**) and (**E**), each frame corresponds to 1 min interval. Scale bar, 5 μm. In (**C**) and (**H**), bold curves correspond to the mean and error bars to the standard deviation. In (**C**, **D**) and (**F**, **G**), lines correspond to mean and standard deviation. p-values were calculated using Mann–Whitney U test. Data from n cells was collected from at least three independent experiments.

The online version of this article includes the following source data and figure supplement(s) for figure 3:

**Source data 1.** Numerical data used for *Figure 3B,C,D,F,G, and H*.

**Figure supplement 1.** Time-lapse images of *dis1Δ* and *cut7-24* dis1Δ cells expressing mCherry-Atb2 (tubulin) and Cdc13-GFP (cyclin B) at 37°C.

**Figure supplement 2.** Time-lapse images of *sad1-1* cells expressing mCherry-Atb2 (tubulin) and Sid1-GFP (SPBs), Klp9-GFP or Cut7-GFP at 37°C.

*Figure 3 continued on next page*

*Figure 3 continued*

**Figure supplement 3.** Time-lapse images of wild-type and *klp9Δ* cells expressing mCherry-Atb2 (tubulin) and Cls1-3xGFP (CLASP) at 25˚C.
**Figure supplement 4.** Comparative plot of Cls1-3xGFP intensity throughout anaphase B spindle elongation of wild-type (n = 20) and *klp9Δ* (n = 20).
**Figure supplement 4—source data 1.** Numerical data used for *Figure 3—figure supplement 4*.
**Figure supplement 5.** Time-lapse images of *cut7-24* cells expressing mCherry-Atb2 (tubulin) and Cls1-3xGFP (CLASP) at 37˚C.

Besides, we examined a possible involvement of the kinesin-5 Cut7 and CLASP Cls1 (also called Peg1). Cut7 has been shown to promote spindle elongation like Klp9, even though to a lower extent (*Rincon et al., 2017*; *Yukawa et al., 2019b*), and a dimeric construct of the *X. laevis* kinesin-5 Eg5 promotes microtubule growth in vitro (*Chen et al., 2019*; *Chen and Hancock, 2015*). To test the role of Cut7 we used the *sad1-1* temperature-sensitive mutant to generate monopolar spindles (*Hagan and Yanagida, 1995*). In this mutant, 9.3% of the spindles became bipolar and not all of the monopolar spindles proceeded to anaphase B. We analyzed the spindles that remained monopolar and initiated anaphase B. *Sad1-1* monopolar spindles also assembled long microtubule bundles during anaphase B (*Figure 3E*, *Figure 3—figure supplement 2*), which grew with a similar velocity and to a similar maximum length as the anaphase B bundles in the *cut7-24* mutant (*Figure 3C,D,F,G*, *Figure 3—source data 1*). Thus, *sad1-1* monopolar spindles can also be used to study microtubule dynamics during anaphase B. Deletion of *cut7* was performed in the background of *pkl1* (kinesin-14) deletion, since the deletion of the kinesin-5 alone is lethal (*Olmsted et al., 2014*; *Rincon et al., 2017*; *Syrovatkina and Tran, 2015*; *Yukawa et al., 2018*). Unlike the deletion of *klp9* or *dis1*, the deletion of *cut7* (*cut7Δpkl1Δ*) led to a slight increase of the growth velocity (*Figure 3F, H*) and no significant change of maximum bundle length (*Figure 3G*, *Figure 3—source data 1*, *Figure 3H*). The modest acceleration of bundle growth seemed to be a result of the absence of Cut7 and not Pkl1, since the deletion of *pkl1* alone did not affect growth velocity significantly (*Figure 3F*, *Figure 3—source data 1*). Furthermore, we investigated the localization of Cut7-GFP in the *sad1-1* mutant. While, Klp9-GFP was detected at the tip of microtubule bundles, Cut7-GFP localized only to the unseparated spindle poles (*Figure 3—figure supplement 2*). Thus, even though both motors have been reported to be plus-end directed bipolar kinesins, and both localize to spindle microtubules in a bipolar spindle (*Fu et al., 2009*; *Hagan and Yanagida, 1992*), only Klp9 tracks the tip of the microtubule bundles in monopolar spindles, and promotes their growth during anaphase B. Cut7, like other kinesin-5 members, has been shown to move bidirectionally along microtubule tracks (*Edamatsu, 2014*; *Roostalu et al., 2011*). This bidirectional motility may be the basis for the differential localization of Cut7. Accordingly, Klp9 displays a much stronger affinity for the spindle midzone, where microtubules terminate with their plus-ends, in bipolar spindles than Cut7, which also localizes to regions closer to spindle poles (*Loncar et al., 2020*; *Yukawa et al., 2020*).

Last, we examined if the phenotype upon *klp9* deletion could stem from an interaction with the fission yeast CLASP Cls1. Cls1 localizes to the spindle midzone during anaphase B and promotes microtubule rescues, thus preventing microtubules from depolymerizing back to the spindle poles and preserving spindle stability (*Bratman and Chang, 2007*). Since deletion of *cls1* is lethal, we tested a potential interaction with Klp9 during anaphase B by expressing Cls1-3xGFP in wild-type and *klp9* deleted cells. In bipolar spindles, Cls1-3xGFP localized to the spindle midzone in presence or absence of Klp9 (*Figure 3—figure supplement 3*) with similar intensities throughout anaphase B (*Figure 3—figure supplement 4*, *Figure 3—figure supplement 4—source data 1*). Moreover, in monopolar *cut7-24* spindles, Cls1-3xGFP did not localize to the bundle tip but was rather spread all along the bundle (*Figure 3—figure supplement 5*). Therefore, we conclude that Cls1 is not involved in the underlying mechanism of Klp9-mediated microtubule growth during anaphase B.

## Dis1 regulates the recruitment of Klp9 to the anaphase B spindle

The XMAP215 family member Dis1 emerged as a possible candidate that acts in the pathway with Klp9. To probe this, we examined its role during anaphase B spindle elongation in bipolar spindles. Similar to the deletion of *klp9*, the deletion of *dis1* decreased the spindle elongation velocity (*Figure 4A,B*, *Figure 4—source data 1*). Simultaneous deletion of both proteins did not display an additive effect, and decreased spindle elongation velocity to not significantly different values as the deletion of individual proteins (*Figure 4A,B*, *Figure 4—source data 1*), suggesting that Klp9 and

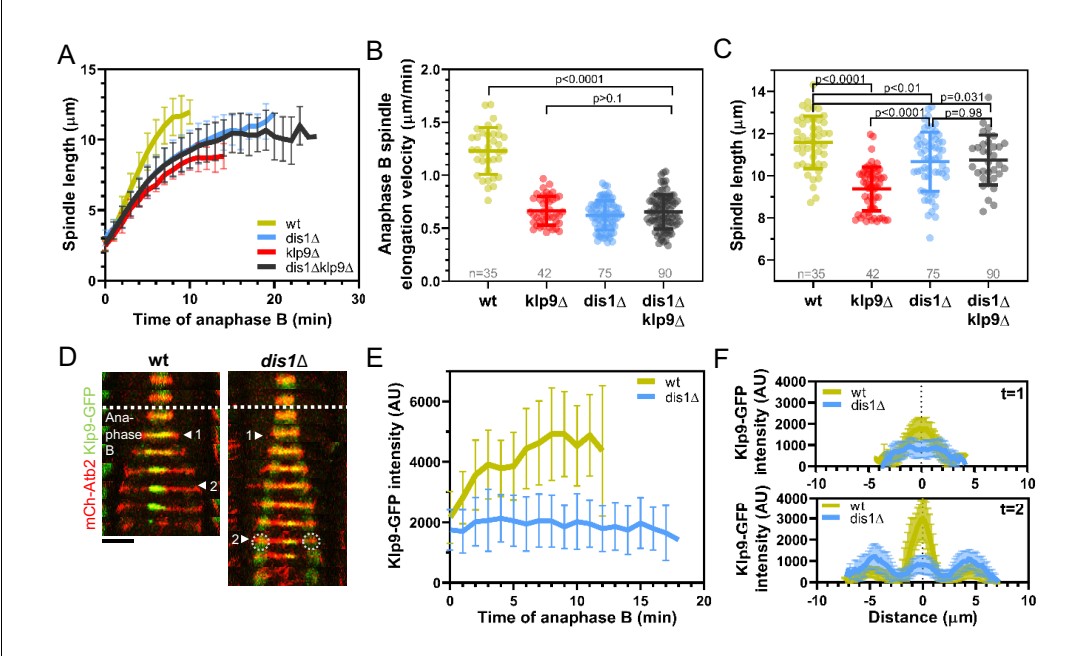

**Figure 4.** *Dis1* deletion impairs Klp9 recruitment to the anaphase B spindle. (**A**) Comparative plot of anaphase B spindle dynamics of wild-type (n = 35), *dis1Δ* (n = 75), *klp9Δ* (n = 38), and d*is1Δklp9Δ* (n = 90) at 25°C. (**B**) Dot plot comparison of spindle elongation velocity in wild-type, *dis1Δ*, *klp9Δ*, and *dis1Δklp9Δ* cells. (**C**) Dot plot comparison of final anaphase B spindle length in wild-type, *dis1Δ*, *klp9Δ*, and *dis1Δklp9Δ* cells. (**D**) Time-lapse images of wild-type and *dis1Δ* cells expressing mCherry-Atb2 (tubulin) and Klp9-GFP at 25°C. Arrowheads 1 and 2 correspond to the time points used for linescan analysis at t = 1 (2 min after anaphase B onset), t = 2 (2 min before spindle disassembly). Circles mark the Klp9 pool that remained in the nucleoplasm. Scale bar, 5 µm. (**E**) Comparative plot of Klp9-GFP intensity throughout anaphase B spindle elongation of wild-type (n = 30) and *dis1Δ* cells (n = 33). (**F**) Intensity spectra obtained by linescan analysis of Klp9-GFP signals along the anaphase B spindle at early (t = 1) and late anaphase (t = 2). In (**A**), (**E**), and (**F**), bold curves correspond to the mean and error bars to the standard deviation. In (**B**) and (**C**), lines correspond to mean and standard deviation. Data from n cells was collected from at least three independent experiments.

The online version of this article includes the following source data and figure supplement(s) for figure 4:

**Source data 1.** Numerical data used for *Figure 4A,B,C,E, and F*.

**Source data 2.** Numerical data used for *Figure 4—figure supplements 1*, *2*, *4*, *5*, and *6*.

**Figure supplement 1.** Dot plot comparison of cell length (µm) of wild-type, *klp9Δ*, and dis1Δ cells at mitosis onset.

**Figure supplement 2.** Dis1 recruitment to the anaphase B spindle is not impaired upon *klp9* deletion.

**Figure supplement 3.** Time-lapse images of *cut7-24* cells expressing mCherry-Atb2 (tubulin) and Dis1-EGFP at 37°C.

**Figure supplement 4.** Dot plot comparison of Klp9-GFP intensity in the nucleoplasm at mitosis onset and mean Klp9-GFP intensity at the spindle midzone during anaphase B in wild-type and *dis1Δ* cells.

**Figure supplement 5.** Comparative plot of the relative Klp9 concentration at the spindle midzone (Klp9-GFP intensity normalized with the mCherry-Atb2 intensity) throughout anaphase B spindle elongation of wild-type (n = 29) and *dis1Δ* (n = 28).

**Figure supplement 6.** Mild overexpression of *dis1* increases Klp9-GFP levels at the spindle midzone during anaphase B.

Dis1 act in the same pathway. Furthermore, final spindle length is reduced in all three mutants (*Figure 4C*, *Figure 4—source data 1*), further indicating an effect of Dis1 and Klp9 on microtubule growth in bipolar spindles.

The longer spindles in *dis1* deleted cells as compared to *klp9Δ* cells may be a result of an increased cell size upon *dis1* deletion (*Figure 4—figure supplement 1*, *Figure 4—source data 2*) due to a slower growth rate in this mutant.

Dis1 was previously shown to be a microtubule polymerase (*Matsuo et al., 2016*). It was thus tempting to think that Klp9 transports Dis1 to the plus-ends of microtubules. However, during anaphase B, Dis1-EGFP localization to spindle poles and the lateral spindle microtubule lattice of bipolar spindles, and the Dis1-EGFP intensity on the spindle was not altered in the absence of Klp9 (*Figure 4—figure supplement 2*, *Figure 4—source data 2*). Moreover, in monopolar spindles, Dis1-EGFP localized to the unseparated spindle poles and disperse along the microtubule bundle (*Figure 4—figure supplement 3*, *Figure 4—source data 2*), but did not accumulate at the bundle tip.

These results suggest that Klp9 does neither regulate the recruitment of Dis1 nor its localization along the anaphase B spindle in bipolar or monopolar spindles. Dis1, thus, cannot be directly responsible for the reduced microtubule growth rate upon *klp9* deletion in monopolar spindles.

Therefore, we wondered if Dis1 could recruit Klp9. Indeed, we observed that the Klp9-GFP signal at the spindle midzone was strongly diminished upon *dis1* deletion (*Figure 4D*, *Figure 4—source data 1*). While in the presence of Dis1, the intensity of Klp9-GFP increased with progressing spindle elongation, until a plateau was reached in late anaphase, the intensity remained low in *dis1Δ* cells (*Figure 4E*, *Figure 4—source data 1*). This is not a consequence of overall lower Klp9 levels, since the total intensity of Klp9-GFP (measured at mitosis onset in the nucleoplasm, where Klp9 is localized before anaphase B onset) was not significantly altered (*Figure 4—figure supplement 4*, *Figure 4—source data 2*). Intensity profiles along the spindle at early and late anaphase B showed that Klp9-GFP still localized preferentially to the midzone in absence of Dis1, but with reduced intensity (*Figure 4F*, *Figure 4—source data 1*). The increased intensity of Klp9-GFP close to the spindle poles in late anaphase B in *dis1Δ* (*Figure 4F*, t = 2) corresponded to the pool of Klp9-GFP that remained in the nucleoplasm and was thus not recruited to the spindle (*Figure 4D*, circles). To further exclude the possibility, that the decreased Klp9-GFP intensity could be a result of a decreased spindle microtubule number, and thus fewer binding-sites for Klp9 upon *dis1* deletion, we normalized the Klp9-GFP intensity with the mCherry-Atb2 intensity. This relative Klp9 concentration at the midzone increased in wild-type cells, according to the increasing Klp9-GFP intensity, but remained low in *dis1Δ* cells (*Figure 4—figure supplement 5*, *Figure 4—source data 2*). Hence, the impaired localization of Klp9-GFP did not stem from differences in microtubule number.

Moreover, mild overexpression of *dis1* by inserting the thiamine repressible nmt promoter *pnmt81* upstream of the *dis1* open-reading frame increased the intensity of Klp9-GFP throughout anaphase B at the midzone, while the total Klp9-GFP intensity was not significantly different (*Figure 4—figure supplement 6*, *Figure 4—source data 2*). Thus, the microtubule polymerase Dis1 regulates the recruitment of the kinesin-6 motor Klp9 to the anaphase B spindle in a dose-dependent manner.

## Dis1 regulates the recruitment of Klp9 in a dephosphorylation-dependent manner

Klp9 localization during mitosis is regulated in a phosphorylation-dependent manner (*Figure 5A*). The motor is phosphorylated at mitosis onset by Cdc2, homolog of Cdk1, and dephosphorylated at the metaphase-to-anaphase B transition by Clp1, homolog of Cdc14 (*Fu et al., 2009*). Dephosphorylation was proposed to be required for Klp9 recruitment to the anaphase B spindle (*Fu et al., 2009*). The localization of Dis1 is also regulated via phosphorylation/dephosphorylation at Cdc2 phosphosites (*Aoki et al., 2006*; *Figure 5A*). Cdc2-mediated phosphorylation of Dis1 promotes its localization to kinetochores during pro- and metaphase (*Aoki et al., 2006*), where it is involved in sister chromatid separation (*Nabeshima et al., 1995*; *Ohkura et al., 1988*). Dephosphorylation of the Cdc2 phosphosites by an unknown phosphatase at the metaphase-to-anaphase B transition results in Dis1 relocalization to the lateral microtubule lattice of anaphase B spindles (*Figure 5A*; *Aoki et al., 2006*). At this location, Dis1 may be required for bundling parallel microtubules (*Roque et al., 2010*). Accordingly, a phospho-mimetic Dis1 mutant cannot be detected on spindle microtubules, only at spindle poles, and a phospho-deficient Dis1 version localizes extensively to the parallel microtubule lattice of the mitotic spindle (*Aoki et al., 2006*).

We thus asked if Dis1 is also dephosphorylated by Clp1 and, furthermore, if this dephosphorylation is required for Dis1-mediated Klp9 recruitment.

Upon *clp1* deletion, Dis1-EGFP was only detected at spindle poles, and not on spindle microtubules during anaphase B (*Figure 5B–D*), resulting in a decreased Dis1-EGFP intensity along the spindle (*Figure 5E*, *Figure 5—source data 1*). This localization pattern of Dis1-EGFP was equal to a phospho-mimetic Dis1 mutant (*Aoki et al., 2006*), indicating that in the absence of Clp1, Dis1 remained phosphorylated. Moreover, *clp1* deletion resulted in a significant reduction of the Klp9-mCherry intensity at the spindle midzone (*Figure 5B, D, and F*, *Figure 5—source data 1*). Consequently, the spindle elongation velocity in anaphase B was reduced upon *clp1* deletion (*Figure 5—figure supplement 1*, *Figure 5—source data 2*) as well as the growth of microtubule bundles in monopolar *cut7-24* spindles (*Figure 5—figure supplement 2*, *Figure 5—source data 1*), in agreement with a reduced intensity of Klp9-GFP at the bundle tips (*Figure 5G*).

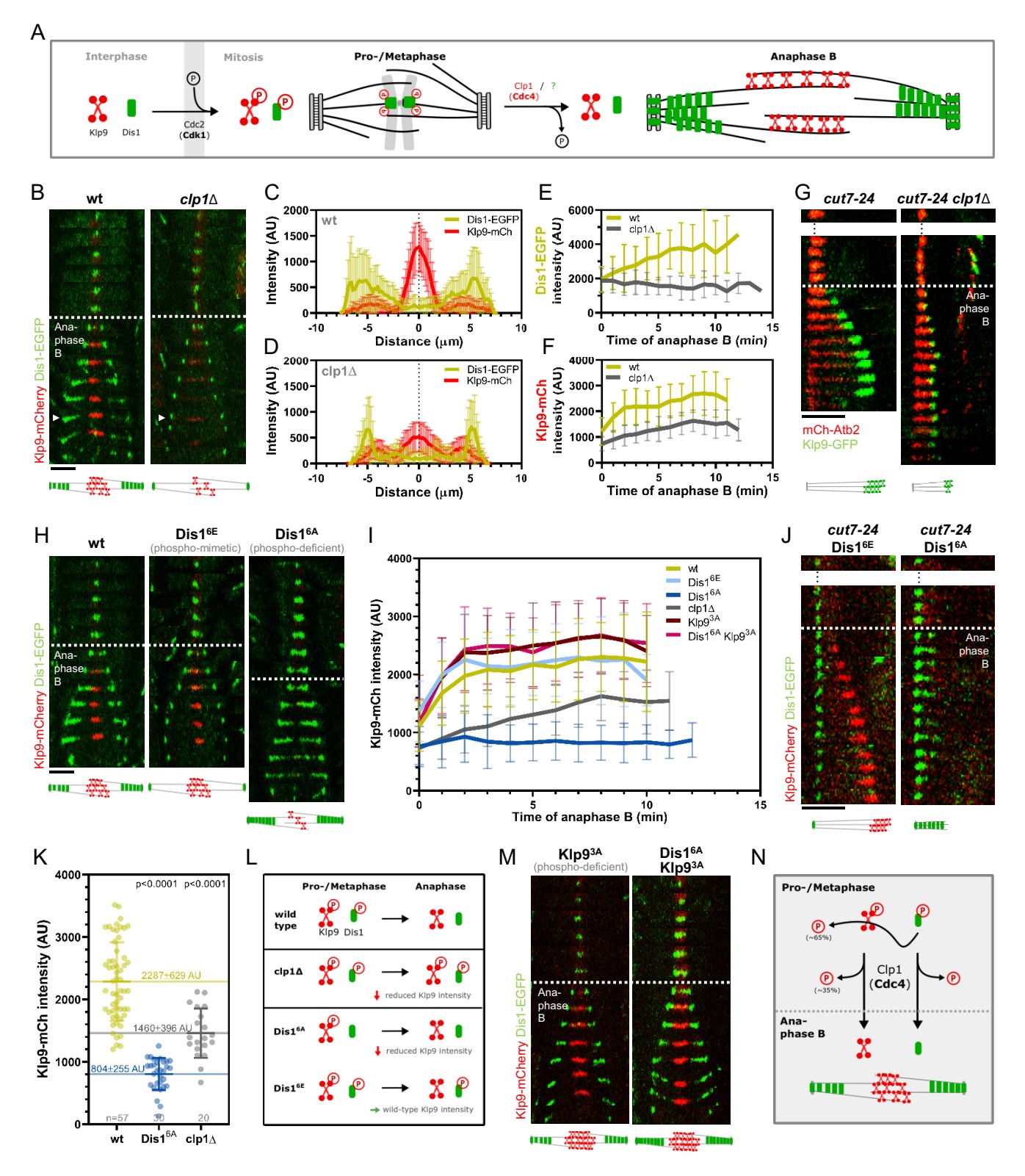

**Figure 5.** Phospho-dependent regulation of Dis1-mediated Klp9 recruitment. (**A**) Model of phosphorylation-dependent localization of Klp9 (red) and Dis1 (green) throughout mitosis mediated by the cyclin-dependent kinase Cdc2 (homolog of Cdk1) and the phosphatase Clp1 (homolog of Cdc14). (**B**) Time-lapse images of wild-type and *clp1Δ* cells expressing Klp9-mCherry and Dis1-EGFP at 25°C. Arrowhead depicts the time point used for linescan analysis (2 min before spindle disassembly). Schemes of anaphase B spindles, illustrating the localization pattern of Klp9 (red) and Dis1 (green) along

*Figure 5 continued on next page*

Figure 5 continued

the spindle (gray). (C) Intensity spectra obtained by linescan analysis of Dis1-EGFP and Klp9-mCherry signals along the anaphase spindle at late anaphase in wild-type cells (n = 30). X = 0 μm equals the cell center. (D) Intensity spectra obtained by linescan analysis of Dis1-EGFP and Klp9-mCherry signals along the anaphase spindle at late anaphase in *clp1Δ* cells (n = 30). (E) Comparative plot of Dis1-EGFP intensity throughout anaphase B spindle elongation of wild-type (n = 30) and *clp1Δ* (n = 30). (F) Comparative plot of Klp9-mCherry intensity throughout anaphase B spindle elongation of wild-type (n = 30) and *clp1Δ* (n = 30). (G) Time-lapse images of *cut7-24*, and *cut7-24 clp1Δ* cells expressing Klp9-GFP and mCherry-Atb2. Time interval corresponds to 1 min. Schemes of monopolar anaphase B spindles, illustrating the localization pattern of Klp9 (green) along the spindle (grey). (H) Time-lapse images of cells expressing wild-type Dis1-GFP, phospho-mimetic Dis1$^{6E}$-GFP or phospho-deficient Dis1$^{6A}$-GFP together with Klp9-mCherry at 25°C. (I) Comparative plot of Klp9-mCherry intensity throughout anaphase B spindle elongation of wild-type cells (n = 30) and cells expressing Dis1$^{6E}$-GFP (n = 30), Dis1$^{6A}$-GFP (n = 30), Klp9$^{3A}$-mCherry (n = 30), Dis1$^{6A}$-EGFP and Klp9$^{SA}$-mCherry (n = 30), and *clp1Δ* cells (n = 20). Schemes of anaphase B spindles, illustrating the localization pattern of Klp9 (red) and Dis1 (green) along the spindle (gray). (J) Time-lapse images of *cut7-24* cells expressing Klp9-mCherry and Dis1$^{6E}$-EGFP or Dis1$^{6A}$-EGFP. Time interval corresponds to 1 min. Schemes of monopolar anaphase B spindles, illustrating the localization pattern of Klp9 (red) and Dis1 (green) along the spindle (gray). (K) Dot plot comparison of the Klp9-mCherry intensity (AU) in wild-type, Dis1$^{6A}$, *clp1Δ* and Dis1$^{6E}$*clp1Δ* cells. Lines correspond to mean and standard deviation. Long lines depict the mean for each cell type. p-values were calculated using Mann-Whitney U test. (L) Summary of the results obtained upon *clp1* deletion and expression of phospho-deficient Dis1$^{6A}$ or phosphoimimetic Dis1$^{6E}$. (M) Time-lapse images of cells expressing phospho-deficient Klp9$^{3A}$-mCherry with wild-type Dis1-GFP, and phospho-deficient Klp9$^{3A}$-mCherry with phospho-deficient Dis1$^{6A}$-EGFP at 25°C. Schemes of anaphase B spindles, illustrating the localization pattern of Klp9 (red) and Dis1 (green) along the spindle (gray). (N) Model of dephosphorylation-dependent regulation of Klp9 and Dis1 localization to the anaphase B spindle. In (B), (H), and (M), each frame corresponds to 2 min interval. Dotted lines denote the transition to anaphase B. Scale bar, 5 μm. In (C–F) and (I), bold curves correspond to the mean and error bars to the standard deviation. Data from n cells was collected from at least three independent experiments. The online version of this article includes the following source data and figure supplement(s) for figure 5:

**Source data 1.** Numerical data used for *Figure 5C,D,E,F,I, and K*.
**Source data 2.** Numerical data used for *Figure 5—figure supplements 1*, *2,* and *4*.
**Figure supplement 1.** Comparative plot of spindle length dynamics during anaphase B and dot plot compairison of anaphase B spindle elongation velocity of wild-type (n = 20) and *clp1Δ* cells (n = 40).
**Figure supplement 2.** Comparative plot of microtubule bundle dynamics in *cut7*-24 (n = 39), *cut7-24 clp1Δ* (n = 35), *and cut7-24 klp9Δ* (n = 40) at 37°C.
**Figure supplement 3.** Time-lapse images of wild-type and *ase1Δ* cells expressing Klp9-mCherry and Dis1-EGFP at 25°C.
**Figure supplement 4.** Comparative plot of Klp9-mCherry intensity throughout anaphase B spindle elongation of wild-type (n = 30) and *ase1Δ* (n = 30).

Thus, both Klp9 and Dis1 are dephosphorylated at the metaphase-to-anaphase B transition by Clp1, allowing Dis1 to bind to the parallel spindle microtubule lattice and Klp9 recruitment to anti-parallel midzone microtubules (*Figure 5L*).

Besides, Clp1 also dephosphorylates Ase1, and dephosphorylation of Ase1 and Klp9 has been proposed to allow their physical interaction, and eventually the recruitment of Klp9 to the spindle by Ase1 (*Fu et al., 2009*). Hence, we examined if the observed reduction of the Klp9-mCherry intensity in *clp1Δ* did not stem from a prevented Klp9-Ase1 interaction. We could not detect mis-localization of Klp9 or Dis1, nor a reduction of the Klp9-mCherry intensity upon *ase1* deletion (*Figure 5—figure supplements 3* and *4*, *Figure 5—source data 2*). Thus, Ase1 does not appear to recruit Klp9 to the spindle upon dephosphorylation.

Our data suggests that this task is performed by Dis1, since *dis1* deletion disrupts Klp9 recruitment. We then asked if Dis1 dephosphorylation by Clp1 at anaphase B onset is required for Klp9 recruitment to the spindle.

To do so, we took advantage of previously generated phospho-mimetic and phospho-deficient mutants of Dis1. The six consensus Cdc2-phosphosites of Dis1 (T279, S293, S300, S551, S556, and S590) were either changed to alanine, to create a phospho-deficient mutant (Dis1$^{6A}$), or to glutamate, to obtain a phospho-mimetic version of Dis1 (Dis1$^{6E}$) (*Aoki et al., 2006*).

If dephosphorylation of Dis1 at anaphase B onset would be necessary for Klp9 recruitment, we would expect the expression of Dis1$^{6E}$-GFP, mimicking the metaphase phosphorylated state, to result in reduced Klp9-mCherry intensities, and expression of Dis1$^{6A}$-GFP, mimicking the anaphase B dephosphorylated state, to lead to unaltered Klp9-mCherry intensities at the spindle midzone. Contrarily, we found that the intensity of Klp9-mCherry was not altered in cells expressing the phospho-mimetic Dis1$^{6E}$-GFP, but was strongly reduced in cells expressing the phospho-deficient Dis1$^{6A}$-GFP compared to the expression of wild-type Dis1-GFP (*Figure 5H, I*, *Figure 5—source data 1*). Accordingly, in *cut7-24* cells, expression of Dis1$^{6E}$-GFP allowed Klp9-mCherry recruitment to the growing microtubule bundle tip, whereas expression of Dis1$^{6A}$-GFP prevented Klp9-mCherry localization to the bundle and subsequently its growth (*Figure 5J*). This indicates that the recruitment of Klp9 by

Dis1 does not require Dis1 dephosphorylation but depends on its phosphorylation: For proper Klp9 recruitment, Dis1 has to be phosphorylated at Cdc2-phosphosites prior to anaphase B onset (*Figure 5L*).

This also suggests that, the reduced Klp9-mCherry intensity upon *clp1* deletion is not a result of the prevented dephosphorylation of Dis1. Two different pathways may regulate Klp9 localization to the anaphase B spindle: one depends on dephosphorylation of Klp9 by Clp1 and the other requires the presence of phosphorylated Dis1 during pro-/metaphase. Indeed, the decrease of the Klp9-mCherry intensity was milder upon *clp1* deletion, than it was upon expression of Dis1[6A] (*Figure 5I, K*, *Figure 5—source data 1*). We noticed that the sum of the Klp9-mCherry intensity in cells expressing Dis1[6A]-GFP (804 ± 255 AU) and *clp1Δ* cells (1460 ± 396 AU) yields a similar value as the intensity in wild-type cells (2287 ± 629 AU) (*Figure 5K*, *Figure 5—source data 1*). Hence, Klp9 recruitment appears to be regulated by two pathways: one relying on phosphorylated Dis1 (~65%) and one on Clp1 (~35%).

Last, we asked how phosphorylated Dis1, present during pro- and metaphase, regulates the recruitment of Klp9 to spindle microtubules at anaphase B onset.

Similar to Clp1, Dis1 may promote Klp9 dephosphorylation. Phosphorylated Dis1 could initiate a pathway at the end of metaphase, which triggers Klp9 dephosphorylation at anaphase B onset, eventually enabling Klp9 binding to the spindle midzone.

To probe this hypothesis, we decided to test whether the expression of a phospho-deficient version of Klp9 could rescue the decreased Klp9 recruitment observed upon expression of phospho-deficient Dis1[6A]-GFP. Previously, three Cdc2-dependent phosphorylation sites of Klp9 (S598, S605, and S611) have been mutated to alanine to obtain phospho-deficient Klp9 (Klp9[3A]) (*Fu et al., 2009*). Indeed, the intensity of Klp9[3A]-mCherry in cells expressing Dis1[6A]-GFP was not significantly different compared to cells expressing wild-type Dis1-GFP and Klp9[3A]-mCherry (*Figure 5I,M*, *Figure 5—source data 1*). Thus, the reduced Klp9 recruitment upon expression of Dis1[6A]-GFP could be rescued by simultaneous expression of Klp9[3A]-mCherry (*Figure 5M*). Hence, Dis1 appears to regulate recruitment of the motor to anaphase B spindles by promoting Klp9 dephosphorylation.

The results at hand suggest the following mechanism for the recruitment of the kinesin-6 Klp9 to anaphase B spindles (*Figure 5N*): Klp9 and Dis1 are phosphorylated at Cdc2-phosphosites during pro- and metaphase. Phosphorylated Dis1, potentially at the end of metaphase, initiates a pathway that regulates dephosphorylation of approximately 65% of the recruited Klp9 pool at anaphase B onset. Dephosphorylation allows the motor to bind to the spindle midzone. The remaining 35% of Klp9 are dephosphorylated by Clp1, which also dephosphorylates Dis1 in order to allow its binding to parallel spindle microtubules during anaphase B.

## Klp9 sets the microtubule growth velocity in vitro

Since Dis1 acts upstream of Klp9, the effect of Klp9 on microtubule polymerization in monopolar spindles may arise from the motor function itself. To probe this hypothesis, we examined the effect of recombinant full-length Klp9 (*Figure 6—figure supplement 1*) on dynamic microtubules in vitro. The functionality of the kinesin-6 was verified using a microtubule gliding assay (*Figure 6A*). Klp9 was immobilized on a glass coverslip via a His6-antibody and red ATTO-647-labeled taxol-stabilized microtubules polymerized from bovine tubulin were added to the flowchamber (*Figure 6A*). Using total internal reflection fluorescence (TIRF) microscopy, microtubule motion was observed. Microtubules moved with an average velocity of 2.4 ± 0.4 μm/min (*Figure 6B,C*, *Figure 6—source data 1*), which was consistent with previously observed Klp9-mediated microtubule gliding velocities (*Yukawa et al., 2019a*).

Subsequently, we examined the effect of Klp9 on dynamic microtubules growing from biotin- and ATTO-647-labeled, GMPCPP-stabilized microtubule seeds (*Figure 6D*). In the presence of 10 μM free tubulin (80% unlabeled, 20% ATTO-488 labeled), the addition of Klp9 increased the growth velocity in a dose-dependent manner from 1.0 ± 0.3 μm/min, in the absence of Klp9, to 1.7 ± 0.4 μm/min, in the presence of 25 nM Klp9 (*Figure 6E,F*, *Figure 6—source data 1*). Moreover, microtubule length, measured just before a growing microtubule underwent catastrophe, increased (*Figure 6G*, *Figure 6—source data 1*) and the catastrophe frequency decreased with increasing Klp9 concentration (*Figure 6H*, *Figure 6—source data 1*). Taken together, the kinesin-6 Klp9 stabilized microtubules and enhanced microtubule growth by increasing the velocity of polymerization and by decreasing the catastrophe frequency. This has similarly been observed for the *S. cerevisiae*

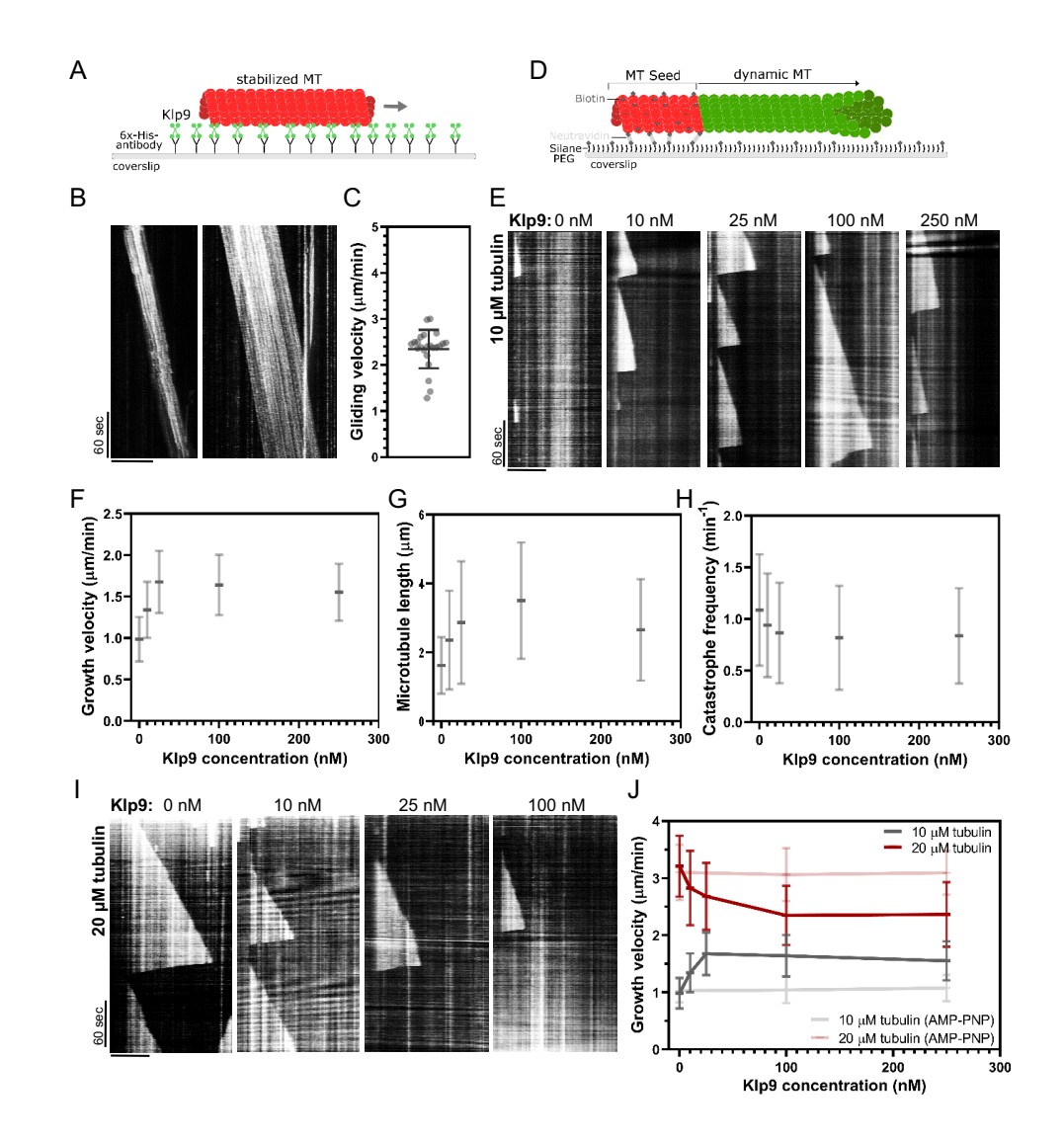

**Figure 6.** Recombinant Klp9 regulates the microtubule growth velocity in vitro. (**A**) Schematic set up of the microtubule gliding assay. His6 antibodies are shown in gray, Klp9 molecules in green, and taxol-stabilized microtubules in red. (**B**) Kymographs of gliding microtubules with the time on the y-axis and space on the x-axis. Scale bar, 10 μm. (**C**) Dot plot of microtubule gliding velocities. (**D**) Schematic of microtubule polymerization assay: GMPCPP-stabilized, ATTO-647-labeled microtubule seeds are shown in red and the dynamic microtubule grown from free tubulin (80% unlabeled, 20% ATTO-488-labeled tubulin) in green. (**E**) Kymographs of dynamic microtubules grown in presence of 10 μM free tubulin and 0, 10, 25, 100, and 250 nM Klp9. Scale bar, 5 μm. (**F**) Microtubule growth velocity (μm/min) shown as a function of the Klp9 concentration (nM); n = 190–325 microtubules per condition. (**G**) Microtubule length (μm) shown as a function of the Klp9 concentration (nM); n = 157–325 microtubules per condition. (**H**) Catastrophe frequency ($\text{min}^{-1}$) shown as a function of the Klp9 concentration (nM); n = 65–112 microtubules per condition. (**I**) Kymographs of dynamic microtubules grown in presence of 20 μM free tubulin (80% unlabeled, 20% ATTO-488-labeled tubulin), and 0, 10, 25, and 100 nM Klp9. Scale bar, 5 μm. (**J**) Microtubule growth velocity (μm/min) measured in the presence of 10 μM or 20 μM free tubulin, and ATP or AMP-PNP, shown as a function of the Klp9 concentration (nM); n = 190–399 microtubules per condition. For (**F–H**) and (**J**), mean values and standard deviations are shown. Data from n microtubules was collected from at least three independent experiments.

The online version of this article includes the following source data and figure supplement(s) for figure 6:

**Source data 1.** Numerical data used for *Figure 6C,F,G,H, and J*.

**Source data 2.** Numerical data used for *Figure 6—figure supplements 2* and *3*.

**Figure supplement 1.** SDS-gel of purified full-length Klp9 (71 kDa) stained with Instant Blue (Euromedex).

**Figure supplement 2.** Microtubule length (μm) measured in the presence of 10 μM or 20 μM free tubulin shown as a function of the Klp9 concentration (nM); n = 190–399 microtubules per condition.

*Figure 6 continued on next page*

*Figure 6 continued*

**Figure supplement 3.** Catastrophe frequency (min$^{-1}$) measured in presence of 10 µM or 20 µM free tubulin shown as a function of the Klp9 concentration (nM); n = 190–399 microtubules per condition.

plus-end directed kinesin Kip2 (*Hibbel et al., 2015*) and *X. laevis* kinesin-5 (*Chen and Hancock, 2015*).

Surprisingly, we found that at a higher tubulin concentration (20 µM), which allowed faster microtubule growth in the control condition (3.21 ± 0.53 µm/min), Klp9 displayed the opposite effect on microtubule growth (*Figure 6I,J*, *Figure 6—source data 1*). With increasing Klp9 concentration, the microtubule growth velocity decreased until it reached a plateau, ranging around 2.4 ± 0.5 µm/min in the presence of 100 nM Klp9 (*Figure 6J*, *Figure 6—source data 1*). Note that this velocity was similar to the Klp9-mediated microtubule gliding velocity (*Figure 6C*). Moreover, the microtubule length decreased with increasing Klp9 concentration (*Figure 6—figure supplement 2*, *Figure 6—source data 2*), and the catastrophe frequency did not change significantly (*Figure 6—figure supplement 3*, *Figure 6—source data 2*).

To probe, if the effect on the microtubule growth velocity was dependent on Klp9's motor activity, we performed the polymerization assays with 0, 100, and 250 nM Klp9 in the presence of the non-hydrolysable ATP analog AMP-PNP. In the presence of 10 µM or 20 µM free tubulin, the microtubule growth velocity was not significantly different in the absence or presence of Klp9 (*Figure 6J*, *Figure 6—source data 1*). Hence, both the Klp9-dependent increase of the microtubule growth velocity when the tubulin concentration is low, and the Klp9-dependent decrease of the microtubule growth velocity when the tubulin concentration is high, require motor activity of the kinesin-6.

The results suggest that the kinesin-6 does not only promote microtubule polymerization it can also decrease the microtubule growth velocity depending on the tubulin concentration. The kinesin-6 may thus have adopted a mechanism that allows the motor to set a definite microtubule growth velocity, close to its walking speed. To our knowledge, such behavior has not yet been described for kinesins or other MAPs.

## Discussion

This study leads to two main conclusions: (i) The localization of the kinesin-6 Klp9 to spindle microtubules at anaphase B onset requires its dephosphorylation at Cdc2-phosphosites, mediated mostly by the XMAP215 family member Dis1 (phosphorylated at Cdc2-phosphosites) and with a smaller proportion by Clp1 (homolog of Cdc14). (ii) Klp9 promotes microtubule polymerization in vivo and in vitro in a dose-dependent manner. Moreover, in vitro at high tubulin concentration, where microtubules grow comparatively fast, an increasing Klp9 concentration caused a decrease of the microtubule growth speed, suggesting that Klp9 acts as a cruise control by setting a distinct microtubule growth velocity.

### Dis1-dependent localization of Klp9 to the anaphase spindle

Members of the XMAP215/Dis1 family have been identified as MAPs that accelerate the rate of microtubule growth from yeast to human (*Al-Bassam et al., 2012*; *Brouhard et al., 2008*; *Charrasse et al., 1998*; *Gard and Kirschner, 1987*; *Kinoshita et al., 2001*; *Matsuo et al., 2016*; *Podolski et al., 2014*; *Tournebize et al., 2000*). Moreover, recently XMAP215 family members have been implicated in the regulation of microtubule nucleation (*Roostalu et al., 2015*; *Thawani et al., 2018*; *Wieczorek et al., 2015*). These functions make XMAP215 proteins essential for proper mitotic spindle functioning, with their knockdown leading to small or disorganized spindles (*Cassimeris and Morabito, 2004*; *Cullen et al., 1999*; *Garcia et al., 2001*; *Gergely et al., 2003*; *Goshima et al., 2005*; *Kronja et al., 2009*; *Matthews et al., 1998*; *Ohkura et al., 1988*; *Reber et al., 2013*; *Severin et al., 2001*; *Tournebize et al., 2000*). We found that Dis1 is moreover crucial due to its function in regulating the recruitment of the mitotic kinesin-6 Klp9. *X. laevis* XMAP215 has been implicated in the recruitment of Cdc2 to spindle microtubules via interaction with cyclin B (*Charrasse et al., 2000*), yet a direct effect on mitotic motors has not been reported so far. In general, the results suggest that XMAP215 proteins may also serve as a general regulatory unit of mitotic spindle composition. How Klp9 is dephosphorylated by the phosphorylated form of Dis1

remains elusive. Dis1, phosphorylated at Cdc2-phosphosites, could mediate the activation or release of a phosphatase. It is possible that Dis1, which is still phosphorylated in late metaphase, activates a phosphatase, which subsequently dephosphorylates Klp9 at anaphase onset. Recently, similar to Dis1, the inhibitor of PP2A protein phosphatases, Sds23, has been shown to act upstream of Klp9 and regulate Klp9 recruitment to the spindle (*Schutt and Moseley, 2020*). Dis1 and Sds23 may act in the same pathway.

Besides, we were wondering about the need for such an unconventional mechanism. Why is Dis1 implicated in the regulation of Klp9 localization to the anaphase B spindle? Previously, XMAP215 has been shown to regulate mitotic spindle length (*Milunovic-Jevtic et al., 2018*; *Reber et al., 2013*). In vitro and in vivo, XMAP215 sets spindle length in a dose-dependent manner, and may thus be a crucial factor for the scaling of spindle length to cell size (*Krüger and Tran, 2020*; *Milunovic-Jevtic et al., 2018*; *Reber et al., 2013*). Similarly, Klp9 sets the speed of anaphase B spindle elongation in a dose-dependent manner (*Krüger et al., 2019*), which allows scaling of anaphase B spindle elongation velocity to spindle length and cell size (*Krüger et al., 2019*). Thus, the mechanism of Dis1 regulating the recruitment of Klp9 to the anaphase spindle could link the regulation of spindle length to the regulation of spindle elongation velocity: an increased amount of Dis1 results in the formation of longer spindles and the recruitment of more Klp9, subsequently allowing faster spindle elongation of the longer spindles. To test this hypothesis, we investigated if Dis1 indeed regulates spindle length scaling, as it has been suggested by varying the XMAP215 amounts in *X. laevis* cell extracts (*Reber et al., 2013*) or in *X. laevis* cells of the same size (*Milunovic-Jevtic et al., 2018*). *Dis1* deletion did not affect spindle length scaling in metaphase (*Figure 7—figure supplement 1*, *Figure 7—figure supplement 1—source data 1*), but strongly impacted the scaling relationship in anaphase B (*Figure 7A*). *Klp9* deletion also diminished the scaling relationship (*Figure 7A*, *Figure 7—source data 1*), either due to its function in microtubule polymerization or sliding, but deletion of *dis1* displayed an even stronger effect, reducing the slope of the linear regression from 0.5 in wild-type cells to 0.21 in *dis1* deleted cells of different sizes (*Figure 7A*, *Figure 7—source data 1*). Thus, Dis1 regulating Klp9 recruitment during anaphase B could indeed link the regulation of spindle length to the regulation of spindle elongation velocity, ensuring that shorter spindles elongate with slower speeds, and longer spindles with higher speeds, as shown previously (*Figure 7B*; *Krüger et al., 2019*). Eventually, this ensures that longer spindles accomplish chromosome segregation within the same time frame as shorter spindles, thus preventing a prolongation of mitosis, which may be harmful for cell viability (*Krüger and Tran, 2020*).

## Regulation of microtubule dynamics by kinesin-6 Klp9

The in vivo and in vitro experiments at hand strongly suggest the kinesin-6 Klp9 to be a crucial regulator of microtubule dynamics. In monopolar spindles, in the presence of Klp9, microtubule bundles grew with a velocity that matches the microtubule growth velocity expected in bipolar spindles. In the absence of Klp9, the growth velocity and length of microtubule bundles was strongly diminished, indicating that the microtubule growth necessary for anaphase B spindle elongation is promoted by the kinesin-6. This is supported by the observation that Klp9 prominently localized to and accumulated at microtubule bundle tips with anaphase B progression. Microtubule plus-end accumulation appears to be an important function for microtubule dynamics regulating kinesins (*Chen and Hancock, 2015*; *Gudimchuk et al., 2013*; *Hibbel et al., 2015*; *Varga et al., 2006*; *Varga et al., 2009*). We have not observed such a strong tip-tracking activity for other anaphase B spindle components in monopolar spindles, namely the microtubule bundler Ase1, the microtubule rescue factor CLASP, the XMAP215 protein Dis1, or the other bipolar sliding motor kinesin-5. Klp9, thus, appears to be crucial for the regulation of microtubule growth during anaphase B.

We acknowledge that the effect of *klp9* deletion is more pronounced in monopolar as compared to bipolar spindles. Spindle length and elongation velocity are diminished to a smaller degree in bipolar spindles. Additional mechanisms regulating microtubule dynamics may be at play in spindles with a midzone. For instance, CLASP is localized strongly to the antiparallel-overlapping microtubules (*Bratman and Chang, 2007*) and promotes microtubule rescue to avoid microtubules depolymerizing back to spindle poles. Due to the absence of antiparallel microtubule bundles, the activity of CLASP may be reduced in monopolar spindles and thus the effect of Klp9 is more pronounced in monopolar spindles.

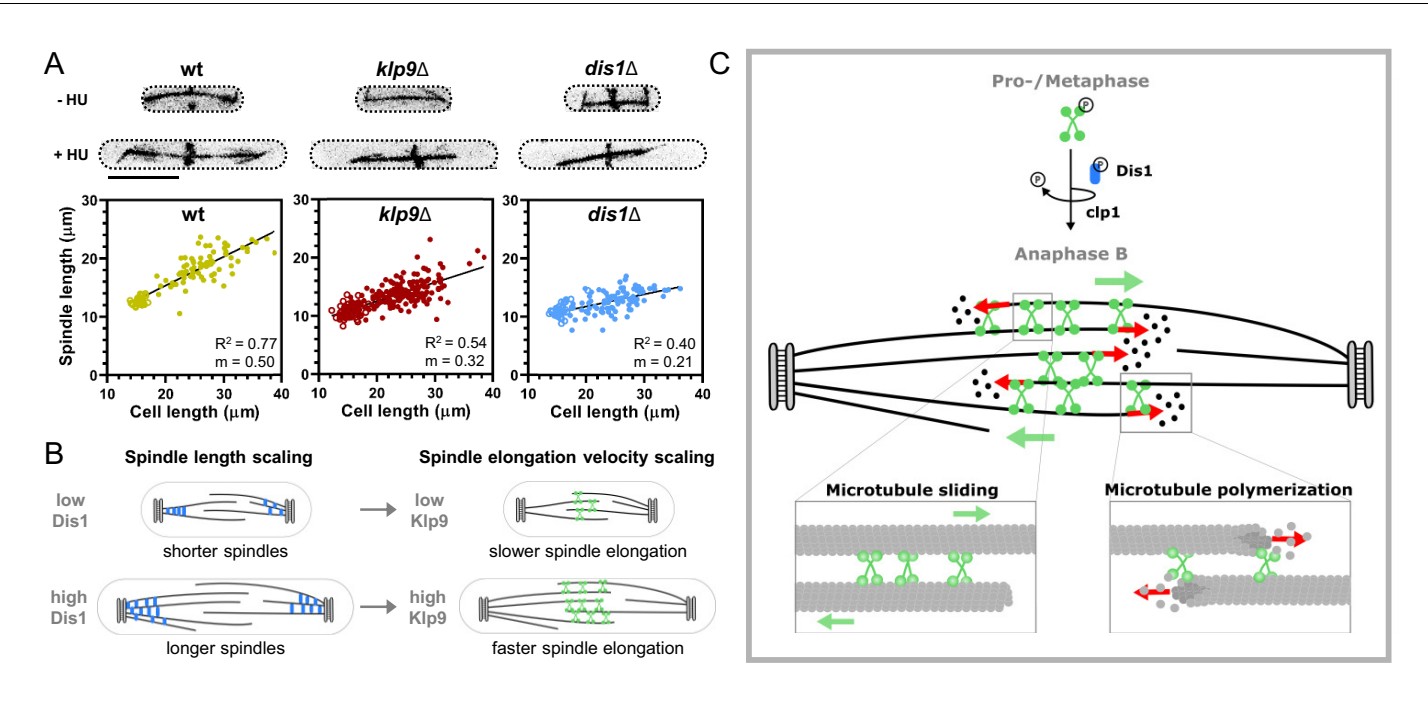

**Figure 7.** Model of Dis1-dependent Klp9 recruitment and Klp9 function during anaphase B spindle elongation. (**A**) Upper: Wild-type, *klp9Δ* and *dis1Δ* cells at the end of anaphase B expressing mCherry-Atb2 treated or not treated with hydroxyurea (HU). Addition of HU allows to block the cells in S-Phase, resulting in increased cell size. Lower: Final anaphase B spindle length plotted against cell length of wild-type, *klp9Δ*, and *dis1Δ* cells. Unfilled circles correspond to cells not treated with HU and filled circles to cells treated with HU. Data was fitted by linear regression, showing the regression coefficient $R^2$ and the slope m. (**B**) Schematic model of the link between spindle length and spindle elongation velocity scaling with cell size. In bigger cells the presence of higher Dis1 (blue) levels gives rise to the assembly of longer spindles as well as the recruitment of higher Klp9 (green) levels to the spindle midzone of anaphase B spindles, which subsequently results in faster spindle elongation of the longer spindles in bigger cells. (**C**) Model of Klp9 recruitment at anaphase onset and its function during anaphase B. Klp9 may promote spindle elongation by generating microtubule sliding forces and regulating the microtubule growth velocity.

The online version of this article includes the following source data and figure supplement(s) for figure 7:

**Source data 1.** Numerical data used for *Figure 7A*.

**Figure supplement 1.** Wild-type, *klp9Δ*, and *dis1Δ* cells at the end of metaphase expressing mCherry-Atb2 treated or not treated with hydroxyurea (HU) and final metaphase spindle length plotted against cell length of wild-type, *klp9Δ*, and *dis1Δ* cells. .

**Figure supplement 1—source data 1.** Numerical data used for *Figure 7—figure supplement 1*.

In vitro reconstitution experiments with recombinant full-length Klp9 provide further evidence for the sliding motors role in regulating microtubule dynamics. With low microtubule growth rates in the control condition, Klp9 increased the growth velocity in a dose-dependent manner. This has similarly been observed for the *X. laevis* kinesin-5 Eg5 (*Chen and Hancock, 2015*). The fact that not only a dimeric, but also a monomeric Eg5 construct promoted microtubule polymerization, suggests that the underlying mechanism is based on Eg5 stabilizing a straight tubulin conformation (*Chen et al., 2019*; *Chen et al., 2016*; *Chen and Hancock, 2015*). In solution, tubulin exhibits a curved conformation, which is not potent for microtubule lattice incorporation (*Ayaz et al., 2012*; *McIntosh et al., 2018*; *Rice et al., 2008*). The motor walks toward the microtubule plus-end, accumulates there by staying bound for time periods that greatly exceed the stepping duration (*Chen and Hancock, 2015*), and straightens newly added tubulin dimers, thus stabilizing lateral tubulin–tubulin interactions and promoting microtubule growth (*Chen et al., 2019*). This differs from the mechanism of microtubule polymerization proposed for members of the XMAP215 family. In contrast to kinesin-5, XMAP215 proteins show high affinity for free tubulin (*Ayaz et al., 2014*; *Brouhard et al., 2008*). XMAP215 TOG domains are proposed to bring tubulin close to the plus-ends, so that it can be incorporated into the microtubule lattice (*Ayaz et al., 2014*; *Brouhard et al., 2008*; *Geyer et al., 2018*). The mode of action of kinesin-5 and XMAP215 may thus differ temporally: XMAP215 promotes tubulin dimer association with the protofilament and kinesin-5 stabilizes this association.

Interestingly, for kinesin-6, we observed that at higher microtubule growth velocities in the control condition, Klp9 displayed a negative effect on microtubule growth. Motor addition decreased the growth velocity in a dose-dependent manner at 20 µM free tubulin. Due to this observation, we hypothesize that the mechanism of microtubule growth regulation is different from the mechanisms that have been suggested for kinesin-5 (*Chen et al., 2019*; *Chen and Hancock, 2015*) or XMAP215 (*Ayaz et al., 2014*; *Brouhard et al., 2008*). First, for kinesin-5 and XMAP215, the addition of recombinant protein to higher tubulin concentrations further increases microtubule growth velocities (*Al-Bassam et al., 2012*; *Brouhard et al., 2008*; *Chen et al., 2019*). Second, the fact that microtubule growth velocities converge with the addition of Klp9 in the presence of low or high tubulin concentrations indicates that Klp9 sets a distinct microtubule growth velocity and suggests a mechanism beyond a stabilizing effect proposed for kinesin-5 (*Chen et al., 2019*) or an enhancement of tubulin dimer addition as regulated by XMAP215 (*Brouhard et al., 2008*). In contrast, to be able to accelerate and decelerate microtubule growth, Klp9 has to promote tubulin dimer addition at microtubule plus-ends, as well as to block it. This is reminiscent of the processive elongator of actin filaments formin (*Courtemanche, 2018*). Formins located at the barbed ends of actin filaments, consisting of a dimer of formin homology two domains (FH2), form a ring-like structure around the actin filament, and can accommodate a 'closed' conformation blocking actin subunit addition, or an 'open' conformation, which allows for subunit addition (*Vavylonis et al., 2006*). The equilibrium between the 'closed' and the 'open' conformation of FH2 dimers subsequently determines the rate of actin elongation (*Gurel et al., 2015*; *Kovar et al., 2006*; *Thompson et al., 2013*). Accordingly, one Klp9 molecule located at the end of each protofilament could promote the association of new tubulin dimers with a certain rate, but block the addition beyond that rate. One motorhead could be bound to the very last tubulin dimer, while the other motorhead binds free tubulin and promotes its incorporation into the microtubule lattice. How the inhibition of tubulin subunits occurs appears to be a complicated question. One possibility is that the binding of Klp9 to free curved tubulin induces a conformational change of tubulin dimers that is much more amenable for lattice incorporation. Like kinesin-5, kinesin-6 could promote tubulin dimer straightening, but unlike kinesin-5, not after lattice incorporation, but before. Even if a free tubulin dimer is in close proximity to the last tubulin dimer in the protofilament where one motorhead of Klp9 is bound, the straightened tubulin dimer bound to the other motorhead will be more potent for addition to the protofilament than a free curved tubulin dimer. Consequently, the velocity of microtubule growth may be set by the stepping rate of the motor. Accordingly, we found the reduced microtubule growth velocity in the presence of Klp9 at high tubulin concentrations to be comparable to the Klp9-mediated microtubule gliding velocity.

Given the observation that in monopolar spindles Klp9 primarily enhances microtubule growth, the question of how relevant the function of Klp9 to reduce the microtubule growth velocity of fast-growing microtubules is in vivo arises. In general, the ability of Klp9 to control the microtubule growth velocity and adjusts its speed, independent of encountering a comparatively fast- or slow-growing microtubule allows the motor to robustly coordinate the microtubule growth velocity to the microtubule sliding velocity. It is possible that in bipolar spindles additional mechanisms that regulate micrototubule dynamics are at play. Thus, also faster growing microtubules may be present in bipolar spindles. The kinesin-6 could reduce the velocity of these microtubules and adjust it to the sliding velocity. On the other hand, the motor can also increase the growth speed of slower growing microtubules and is thus able to adjust the velocity of microtubule growth to the velocity of microtubule sliding of different microtubule populations in the spindle.

Collectively, this work sheds light on the mechanism of Klp9 recruitment and function during anaphase B. Dephosphorylation of Klp9 mediated by Dis1 promotes its localization to the spindle midzone, where it can subsequently promote anaphase B spindle elongation (*Figure 7C*). This mechanism of Klp9 recruitment may link the regulation of mitotic spindle length and spindle elongation velocity to achieve scaling with cell size (*Figure 7B*). Moreover, we show that anaphase B spindle elongation is achieved not only by Klp9 generating microtubule sliding forces, as shown previously (*Fu et al., 2009*; *Rincon et al., 2017*), but also by regulating microtubule polymerization. This makes the kinesin-6 a perfect candidate responsible for the coordination of microtubule sliding and growth (*Figure 7C*). With Klp9 at the spindle midzone, the microtubule sliding velocity and the microtubule growth velocity of plus-ends located at the edge of the midzone are set with similar speeds. Last, the in vitro results suggest an unconventional mechanism of the regulation of microtubule growth. The kinesin-6 appears to not simply enhance tubulin dimer addition at microtubule

plus-ends, as suggested for other microtubule polymerases (*Ayaz et al., 2014*; *Brouhard et al., 2008*; *Chen et al., 2019*; *Chen and Hancock, 2015*), but also block tubulin dimer addition that exceeds a certain rate. Hence, Klp9 may have adopted a mechanism that is comparable to that of formins by being able to promote and block subunit addition to the filament. Eventually, Klp9 can thus set a well-defined microtubule growth velocity of slow- and fast-growing microtubules within the spindle and act as a 'cruise control' of microtubule polymerization.

## Materials and methods

### Production of *Schizosaccharomyces pombe* mutant strains

All used strains are isogenic to wild-type 972 and were obtained from genetic crosses, selected by random spore germination and replica on plates with appropriate drugs or supplements. All strains are listed in the supplementary strain list. Gene deletion and tagging was performed as described previously (*Bähler et al., 1998*).

### Fission yeast culture

All *Schizosaccharomyces pombe* strains were maintained at 25°C and cultured in standard media. Strains were maintained on either YE5S plates or EMM plates supplemented with adenine, leucine, and uracil in the case of the Klp9 shut-off strain, at 25°C and refreshed every third day. In general, cells were transferred into liquid YE5S and imaged the following day at exponential growth.

For overexpression of *dis1*, strains were transferred to liquid EMM supplemented with adenine, leucine, uracil, and thiamine 2 days before imaging. The following day cells were centrifuged at 3000 rpm for 5 min, washed three times with $H_2O$, and resuspended in EMM supplemented with adenine, leucine, and uracil. Following incubation at 25°C for 18–20 hr, cells were imaged.

For shut-off of Klp9, cells were cultured in EMM supplemented with adenine, leucine, and uracil 2 days before imaging. The following day, cells were centrifuged at 3000 rpm for 5 min, resuspended in YE5S, and incubated at 25°C for approximately 20 hr until imaging.

The generation of long cells was achieved by treatment with hydroxyurea (Sigma-Aldrich). Cells were transferred to liquid YE5S at 25°C, and 10 mM 10 mM hydroxyurea was added when cells reached exponential growth. After incubation for 5 hr at 25°C, cells were washed three times with $H_2O$ and resuspended in fresh YE5S. Following 1 hr at 25°C, cells were imaged.

### Live microscopy of *S. pombe* cells

For live-cell imaging, fission yeast cells were mounted on YE5S agarose pads, containing 2% agarose (*Tran et al., 2004*). Temperature-sensitive *cut7-24* and *sad1-1* mutants were incubated at the microscope, which is equipped with a cage incubator (Life Imaging Services), at 37°C for 1 hr before imaging was started. All other strains were imaged at 25°C.

Images were acquired on an inverted Eclipse Ti-E microscope (Nikon) with a spinning disk CSU-22 (Yokogawa), equipped with a Plan Apochromat 100×/1.4 NA objective lens (Nikon), a PIFOC (perfect image focus) objective stepper, a Mad City Lab piezo stage, and an electron-multiplying charge-couple device camera (EMCCD 512 × 512 Evolve, Photometrics).

Stacks of seven planes spaced by 1 μm were acquired for each channel with 100 ms exposure time, binning one and an electronic gain of 300 for both wavelengths. For each time-lapse movie, an image was taken every minute for 90–120 min.

### Construction, expression, and purification of recombinant Klp9

For construction of a plasmid containing full-length Klp9, *klp9* cDNA was amplified from a cDNA library (*Hoffman et al., 2015*) and cloned into pET28a(+) between NcoI and NotI restriction sites, incorporating a C-terminal His6 tag.

For Klp9 purification *E. coli* NiCO21 (DE3), cells were transfected with pET28a-Klp9-His6 and grown in standard 2xYT medium. Following cell lysis and centrifugation, the supernatant was incubated with chitin resin (New England Biolabs) for 45 min at 4°C and eluted on a gravity flow column. The protein solution was then loaded on an HiTrap IMAC HP column (GE Healthcare). Further purification of the Klp9-containing fractions was achieved by gel filtration using a Superdex 200 26/60

column equilibrated with a buffer containing 20 mM HEPES pH 8, 150 mM NaCl, 2 mM MgCl$_2$, 0.2 mM ADP, 0.5 mM TCEP, 0.5 mM PMSF, and 10% glycerol.

## Tubulin purification

Tubulin from fresh bovine brain was purified by three cycles of temperature-dependent assembly and disassembly in BRB80 buffer. Labeling of tubulin with ATTO-488, ATTO-647, or biotin was performed as previously described (*Hyman et al., 1991*).

## In vitro microtubule gliding assays

Taxol-stabilized microtubules were prepared by incubating 56 µM unlabeled tubulin and 14 µM ATTO-647-labeled tubulin in BRB80 (80 mM piperazineN,N[prime]-bis(2-ethanesulfonic acid [PIPES]) pH 6.8, 1 mM EGTA, 1 mM MgCl$_2$) with 20 mM MgCl$_2$, 10 mM GTP, and 25% DMSO in BRB80 at room temperature. The mixture was then diluted 1:111 with 10 µM taxol in BRB80.

The flow chamber was assembled from microscopy slides and glass coverslips cleaned sequentially in acetone in a sonication bath and in ethanol. Then, 0.1 mg/ml Anti-His$_6$ (Sigma-Aldrich) was added to the flow chamber, followed by 30 µM recombinant Klp9. Upon subsequent incubation of the flow chamber with taxol-stabilized microtubules, a buffer containing 0.5% methylcellulose, 125 mM KCl, 12.5 mM MgCl$_2$, 2.5 mM EGTA, 16.6 mM Hepes pH 7.5, 3 mg/ml glucose, 0.1 mg/ml glucose oxidase, 0.03 mg/ml catalase, 8.5 mM ATP, and 0.3% BSA was added and the flow chamber was sealed with vacuum grease. Imaging was performed immediately after at 30°C at the TIRF microscope.

## In vitro microtubule dynamics assays

GMPCPP-stabilized microtubule seeds were prepared by incubating a mixture of 7 µM biotinylated tubulin, 3 µM ATTO-647-labeled tubulin, and 0.5 mM GMPCPP (Interchim) in 1xBRB80 for 1 hr at 37°C. Following, 2.5 µM taxol was added and the mix was incubated for 20 min at room temperature (RT). After centrifugation at 14,000 rpm for 10 min the supernatant was removed and the pellet resuspended in a mix containing 2.5 µM taxol and 0.5 µM GMPCPP in 1xBRB80. The seeds were then snap frozen in liquid nitrogen and stored at −80°C.

Flow chambers were assembled from microscopy slides and glass coverslips with double-sticky tape with three independent lanes, allowing the analysis of three different reaction mixtures in the same flow chamber. Slides and coverslips were cleaned sequentially in acetone in a sonication bath, in ethanol, in 2% Hellmanex (Sigma-Aldrich) and using a plasma-cleaner. Functionalization was achieved by sequential incubation with 1 mg/ml PEG-Silane (30 K, Creative PEGWorks) for microscopy slides or 0.8 mg/ml PEG-Silane and 0.2 mg/ml Biotin-PEG-Silane (5 K, Creative PEGWorks) for coverslips and 0.05 mg/ml NeutrAvidin (Invitrogen) in G-Buffer (2 mM Tris–Cl pH 8, 0.2 mM ATP, 0.5 mM DTT, 0.1 mM CaCl$_2$, 1 mM NaAzide, 6.7 mM Hepes pH 7.5, 50 mM KCl, 5 mM MgCl$_2$, and 1 mM EGTA). Attachment of microtubule seeds to the coverslip was achieved through biotin-neutrAvidin interaction. The reaction mixture with or without Klp9 in G-Buffer contained a tubulin mix (80% unlabeled tubulin, 20% 488-labeled tubulin in BRB80) 0.5% methylcellulose, 50 mM KCl, 5 mM MgCl$_2$, 1 mM EGTA, 6.6 mM Hepes pH 7.5, 3 mg/ml glucose, 0.1 mg/ml glucose oxidase, 0.03 mg/ml Catalase, 6.6 mM Tris–Cl pH 8, 0.65 mM ATP, 1 mM GTP, 11.5 mM DTT, 0.3 mM CaCl$_2$, 3.3 mM NaAzide, 0.1% BSA. The mixture was added to the flow chamber, which was then sealed with vacuum grease, and the dynamic microtubules were imaged immediately at 30°C at the TIRF microscope.

## TIRF microscopy

In vitro reconstitution assays were imaged on an inverted Eclipse Ti-E (Nikon) equipped with Perfect Focus System (PFS 2, Nikon), a CFI Plan Apo TIRF 100×/1.49 N.A oil objective (Nikon), an TIRF-E motorized TIRF illuminator, and an EMCCD Evolve camera (Photometrics). For excitation, 491 nm 42 mW and 642 nm 480 mW (Gataca Systems) were used. The temperature was controlled with a cage incubator (Life Imaging Services).

For microtubule gliding and microtubule dynamics assays, movies of 5 min duration with 1 s interval were acquired.

## Image analysis

Using Metamorph 7.2, maximum projections of each stack were performed for the analysis of spindle dynamics and for presentation and sum projections for intensity measurements.

Spindle and bundle dynamics were examined by the length of the mCherry-Atb2 signal over time. Metaphase and anaphase spindle length refer to the final spindle length of each phase.

Intensity measurements were performed by drawing a region around the area of interest, reading out the average intensity per pixel, subtracting the background, and multiplying this value with the size of the area. Intensity profiles along microtubule bundles or along the bipolar spindle were obtained by drawing a line along the bundle/spindle and reading out the intensity values for each pixel along the line, from which the background intensity was subtracted.

The microtubule gliding velocity in the in vitro assays was revealed by constructing kymographs of the moving microtubules using Metamorph and calculation of the velocity by the microtubule displacement within each 5 min movie.

To analyze microtubule dynamics, parameters of individual microtubules growing from GMPCPP-seed kymographs were constructed with Metamorph. For each microtubule, a kymograph was constructed, and the slope of growth and shrinkage events as well as the microtubule length, corresponding to the length of a microtubule before undergoing catastrophe, was analyzed. The catastrophe frequency was calculated by the number of catastrophe events per microtubule within the 5 min movie divided by the total time the microtubule spent growing.

## Quantification and statistical analysis

Sample number, replicates, and error bars are indicated in the figure legends. All statistical analysis and regression analysis were performed using GraphPad Prism 7. Corresponding details for statistical tests and p-values are included in the figures and/or figure legends.

## Acknowledgements

We thank Manuel Lera Ramirez and Ana Loncar for discussions and critical reading of the manuscript. We thank Moutse Ranaivoson for help with the Klp9 purification. We thank Vincent Fraisier and Lucy Sengmanivong for the maintenance of microscopes at the PICT-IBiSA Imaging facility (Institut Curie), a member of the France-BioImaging national research infrastructure. We thank the Japan National BioResource Project – Yeast Genetic Resource Center (Osaka City University, Osaka University and Hiroshima University) for providing strains. We thank Chris Norbury (Oxford University) for generously providing the *S. pombe* cDNA library.

## Additional information

### Funding

| Funder | Grant reference number | Author |
|---|---|---|
| Ministère de l'enseignement supérieur et de la recherche | | Lara Katharina Krüger |
| Fondation ARC pour la Recherche sur le Cancer | | Lara Katharina Krüger Phong T Tran |
| Ligue Contre le Cancer | | Phong T Tran |
| INCA | | Phong T Tran |
| European Research Council | Grant 771599 ICEBERG | Manuel Théry |

The funders had no role in study design, data collection and interpretation, or the decision to submit the work for publication.

### Author contributions

Lara Katharina Krüger, Conceptualization, Data curation, Formal analysis, Validation, Investigation, Visualization, Methodology, Writing - original draft, Writing - review and editing; Matthieu Gélin, Investigation, Methodology; Liang Ji, Resources; Carlos Kikuti, Methodology, Writing - review and

editing; Anne Houdusse, Resources, Writing - review and editing; Manuel Théry, Laurent Blanchoin, Resources, Supervision, Writing - review and editing; Phong T Tran, Resources, Supervision, Funding acquisition, Project administration, Writing - review and editing

### Author ORCIDs
Lara Katharina Krüger  https://orcid.org/0000-0002-0439-951X
Anne Houdusse  http://orcid.org/0000-0002-8566-0336
Laurent Blanchoin  http://orcid.org/0000-0001-8146-9254
Phong T Tran  https://orcid.org/0000-0002-2410-2277

### Decision letter and Author response
Decision letter https://doi.org/10.7554/eLife.67489.sa1
Author response https://doi.org/10.7554/eLife.67489.sa2

## Additional files

### Supplementary files
• Supplementary file 1. List of *S. pombe* strains used in this study.

• Transparent reporting form

### Data availability
A source data file for all data sets used in Figure 1–7 has been provided.

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
