## [Decision Letter]

**Acceptance summary:**

This study investigates the mechanism of spindle elongation in the model system fission yeast using quantitative live cell imaging and genetic manipulations. The authors present compelling evidence for the kinesin Klp9 being important for the control of both microtubule sliding and microtubule growth during anaphase B, providing new insight into the mechanism coordinating both processes during cell division.

**Decision letter after peer review:**

Thank you for submitting your article "Kinesin-6 Klp9 orchestrates spindle elongation by regulating microtubule sliding and growth" for consideration by *eLife*. Your article has been reviewed by 3 peer reviewers, one of whom is a member of our Board of Reviewing Editors, and the evaluation has been overseen by Anna Akhmanova as the Senior Editor. The following individual involved in review of your submission has agreed to reveal their identity: Stefan Westermann (Reviewer #3).

Essential revisions:

1. Results, line 130: A major finding of the study seems to be that microtubule growth speed in anaphase does not depend on spindle bipolarization. Does this imply that growth speed depends on microtubule length – in which case one might expect that Klp9 accumulation at the ends of monopolar spindle bundles is similar to accumulation of Klp9 at the ends of bipolar microtubule overlaps (but this does not seem to be the case) – or growth speed could instead simply depend on time of mitosis (possibly depending on the kinetics of dephosphorylations)? If data are already available, they could be shown to answer these questions. Otherwise, adding a discussion would be considered appropriate.

2. Results, line 253 and following: The reviewers consider it important that the experiments performed with the sad1-1 allele need clarification. It is a point mutation located in the SUN domain that is very sensitive to temperatures above 32 °C. The experiments are carried out at 37 °C, so presumably the protein disappears completely and this blocks the insertion of SPB into the nuclear envelope. Monopolar spindles formation is likely to be observed due to the only insertion of the mother SPB in those cells where some Sad1.1 molecules still persist. It is important to know the percentage of monopolar spindles in a sad1.1 setting observed in the experimental conditions where the experiments have been carried out to see if sad1.1 is a robust tool to explore monopolar spindles formation at 37C. On the other hand, the Sad1 and the SPB insertion itself could affect the spindle elongation mechanisms during anaphase. Can the authors rule out this possibility?

3. The section on phospho-regulation of Klp9 and Dis1 is confusing (starting around line 345). The concern here is that the interpretation is entirely based on localization data and previously reported effects of phospho-alleles, while more direct means to follow the phosphorylation state of the respective proteins (e.g. shift on an SDS- or Phos-tag gel etc,) and how they are influenced by other alleles are not used. The microscopy and the genetic combination of alleles may simply reach limits here in terms of what can be learned mechanistically. A more satisfying explanation for the relationship between Dis1 and Klp9 would require additional biochemical methods. Some questions that don't appear fully answered are: Is Clp1 really responsible for dephosphorylation of Dis1? Is it really the phosphorylation state of Klp9 that is influenced by the Dis1-6A mutation?

Minimally, this section should be re-written to make it more accessible for the reader and more clearly explain the proposed mechanism. Ideally, a few key points should be tested experimentally: e.g. Phosphorylation state of Klp9 in different mutants by western blotting.

4. Related to the previous point, it was considered a pity that the authors went away from the "half-spindle" assay in the later part of the paper (after Figure 3). Maybe some of the regulatory aspects would become more clear in this setting: Is Clp1 required for polymerization of the half spindle? What about the recruitment of Klp9 to the tip of the bundle? Is that impaired in Dis1-6A, is it increased in the Klp9-3A mutant, etc..? The authors could use this simplified setting more. It seems that the endpoint of the mechanism, presumably involving a dephosphorylation cascade that originates at Dis1, is really Klp9. The observation that Klp9-3A can override the Dis1-6A phenotype would support this notion. Therefore a closer investigation of Klp9-3A or -3E in this setting could be informative.

5. Results, Figure 6. Compared to the experiments performed in cells, the in vitro work appears somewhat preliminary.

(a) Please show a Coomassie gel of purified Klp9.

Additional experiments may be worth considering to answer the following 2 points:

(b) Is Klp9 processive, does it accumulate at microtubule ends? Is its motor activity required for its effect on microtubule dynamics? Please use fluorescently labelled Klp9 to visualize it's localization/motility?

(c) Is Klp9 soluble at the higher tubulin concentration? Could it be that the effect of slowing down microtubule growth could be due protein insolubility (which could easily be seen when fluorescently labelled Klp9 is used)

The following issue should be addressed in the Discussion:

(d) How relevant is the observed behavior at 20 μm tubulin where the microtubule growth speed is significantly faster as in cells? In cells Klp9 clearly only accelerates microtubule growth speed according to the data presented in this study.

*Reviewer 1:*

The authors investigate the roles of kinesin-6 Klp9 and the microtubule polymerase Dis1 for anaphase spindle elongation in fission yeast. Using live cell fluorescence microscopy imaging of bipolar and artificially monopolarized spindles, they find that in cells Klp9 promotes microtubule growth in anaphase. recruitment of Klp9 to microtubules depends on Klp9 dephosphorylation that in turn depends on cdc14 phosphatase Clp1 and in a parallel pathway on Dis1 that itself needs to be phosphorylated in pro/metaphase and is then dephosphorylated in anaphase. The authors also show first experiments investigating how purified Klp9 affects microtubule dynamics in vitro (in the presence of purified tubulin), showing an activity that agrees at low tubulin concentrations with observations in cells, but that differs at higher tubulin concentrations.

Experiments in living cells are performed very carefully and comprehensively, quantitative analysis is performed to a high standard and conclusions are supported by the data. A limitation is that the mechanism by which Dis1 promotes dephosphorylation of Klp9 in a phosphorylation-dependent manner in cells, thereby essentially regulating its function for spindle elongation, remains unclear. Interesting is also the first observation of an unusual effect of purified Klp9 on the dynamic properties of microtubule dynamics, even if several mechanistic questions remain open.

*Reviewer 2:*

In this article, Krüger et al. identified a new role for Kinesin-6 using interesting tools to explore microtubule dynamics during anaphase B in fission yeast, for example, the monopolar spindles generated by the cut7-24 and sad1-1 mutations. Interestingly, they found that kinesin-6 (klp9 in fission yeast) promotes microtubule polymerization in vivo in a dose-dependent manner. The authors suggest that Klp9 could act as a cruise control for microtubule growth during anaphase B. They also found that the XMAP215 family member, Dis1, regulates recruitment of Klp9 to spindle microtubules in a dephosphorylation-dependent manner. In my opinion, these data add important information about a new role for kinesin-6 and its regulation throughout the cell cycle in *S. pombe* that are of interest to cell biologists in general.

Experiments performed with the sad1-1 allele need clarification. It is a point mutation located in the SUN domain that is very sensitive to temperatures above 32 °C. The experiments are carried out at 37 °C, so presumably the protein disappears completely and this blocks the insertion of SPB into the nuclear envelope. Monopolar spindles formation is likely to be observed due to the only insertion of the mother SPB in those cells where some Sad1.1 molecules still persist. It is important to know the percentage of monopolar spindles in a sad1.1 setting observed in the experimental conditions where the experiments have been carried out to see if sad1.1 is a robust tool to explore monopolar spindles formation at 37C. On the other hand, I think the removal of the Sad1 and the SPB insertion itself could affect the spindle elongation mechanisms during anaphase and it would be good if the authors could rule out this possibility.

*Reviewer 3:*

In this manuscript Krüger, Tran and colleagues investigate molecular mechanisms of anaphase spindle elongation in the model system fission yeast, using a combination of genetics and live cell microscopy, complemented by experiments with the purified motor Klp9 in vitro. In particular the authors are interested in understanding the coordination between midzone sliding and microtubule polymerization during anaphase elongation, which is required to maintain a stable midzone length. They use a clever genetic trick to generate monopolar half-spindles in fission yeast, which allows one to follow polymerization events usually masked in the midzone bundle. With this, they identify the Kinesin-6 Klp9 and the XMAP-215 protein Dis1 as key factors promoting microtubule polymerization in the half-spindle. They go on to show that these proteins are regulated by the Cdc14 phosphatase Clp1 and that Klp9 association with the midzone requires prior phosphorylation/dephosphorylation of Dis1 on Cdc2 consensus-sites. Finally, the authors show that Klp9 can either promote or attenuate microtubule polymerization in vitro, depending on the concentration of free tubulin.

Overall, the authors tackle an interesting problem, the execution of the genetic and imaging experiments, the display of the data and the writing/discussion, all follow a very high standard.

The section on phospho-regulation of Klp9 and Dis1 is confusing. My overall concern here is that the interpretation is entirely based on localization data and previously reported effects of phospho-alleles, while more direct means to follow the phosphorylation state of the respective proteins (e.g. shift on an SDS- or Phos-tag gel etc,) and how they are influenced by other alleles are not used. The microscopy and the genetic combination of alleles may simply reach limits here in terms of what can be learned mechanistically. A more satisfying explanation for the relationship between Dis1 and Klp9 would require additional biochemical methods. Some questions that don't appear fully answered to me are: Is Clp1 really responsible for dephosphorylation of Dis1? Is it really the phosphorylation state of Klp9 that is influenced by the Dis1-6A mutation?

I found it a bit of a pity, that the authors went away from the "half-spindle" assay in the later part of the paper (after Figure 3). Maybe some of the regulatory aspects would become more clear in this setting: Is Clp1 required for polymerization of the half spindle? What about the recruitment of Klp9 to the tip of the bundle? Is that impaired in Dis1-6A, is it increased in the Klp9-3A mutant, etc..?. The authors could use this simplified setting more. It seems that the endpoint of the mechanism, presumably involving a dephosphorylation cascade that originates at Dis1, is really Klp9. The observation that Klp9-3A can override the Dis1-6A phenotype would support this notion. Therefore a closer investigation of Klp9-3A or -3E in this setting could be informative.

---

## [Author Response]

Essential revisions:1. Results, line 130: A major finding of the study seems to be that microtubule growth speed in anaphase does not depend on spindle bipolarization. Does this imply that growth speed depends on microtubule length – in which case one might expect that Klp9 accumulation at the ends of monopolar spindle bundles is similar to accumulation of Klp9 at the ends of bipolar microtubule overlaps (but this does not seem to be the case) – or growth speed could instead simply depend on time of mitosis (possibly depending on the kinetics of dephosphorylations)? If data are already available, they could be shown to answer these questions. Otherwise, adding a discussion would be considered appropriate.

We think that the accumulation of Klp9 on microtubule bundles of monopolar spindle is similar to the accumulation at the spindle midzone of bipolar spindles. In both cases the Klp9 intensity increases with progression throughout anaphase B (Figure 1G, 4E) and correlates with bundle (Figure 1H) or spindle length (Krüger et al., 2019). Moreover, *klp9* deletion strongly reduces microtubule growth in monopolar spindles and diminishes the spindle elongation velocity and spindle length in bipolar spindles. Admittedly, the effect on spindle length in bipolar spindles is less dramatic as compared to monopolar spindles. We believe that an additional pathway of microtubule dynamics regulation is at play in bipolar spindles, which is inactive in monopolar spindles. Thus, this pathway presumably relies on the spindle midzone. A possible candidate involved in this pathway may be CLASP (Cls1), which is specifically recruited to antiparallel overlaps (Bratman and Chang, 2007) and induces microtubule rescues (Al-Bassam et al., 2010). We have added this hypothesis on the difference between monopolar and bipolar spindle to the Discussion section. However, even though despite the presence of an additional mechanism, that prevents microtubules from completely depolymerizing back to spindle poles by CLASP, Klp9 may still be crucial in bipolar spindles, as the motor is able to adjust the microtubule growth velocity to the microtubule sliding velocity.

2. Results, line 253 and following: The reviewers consider it important that the experiments performed with the sad1-1 allele need clarification. It is a point mutation located in the SUN domain that is very sensitive to temperatures above 32 °C. The experiments are carried out at 37 °C, so presumably the protein disappears completely and this blocks the insertion of SPB into the nuclear envelope. Monopolar spindles formation is likely to be observed due to the only insertion of the mother SPB in those cells where some Sad1.1 molecules still persist. It is important to know the percentage of monopolar spindles in a sad1.1 setting observed in the experimental conditions where the experiments have been carried out to see if sad1.1 is a robust tool to explore monopolar spindles formation at 37C. On the other hand, the Sad1 and the SPB insertion itself could affect the spindle elongation mechanisms during anaphase. Can the authors rule out this possibility?

In the *sad1.1* mutant 18 out of 193 (9.3%) analyzed spindles became bipolar, often after forming transient monopolar spindles and, moreover, not all monopolar spindles proceed to anaphase B. We have added this information in the Results section. We agree with the concern, that the defective SPB insertion into the NE could have an effect on the regulation of microtubule dynamics. However, due to the fact that the growth velocity and the length of microtubule bundles was similar in the *sad1.1* mutant as compared to the *cut7-24* mutant, we believe that the number of SPBs out of which the microtubules are nucleated is not important for the regulation of microtubule growth during anaphase B.

3. The section on phospho-regulation of Klp9 and Dis1 is confusing (starting around line 345). The concern here is that the interpretation is entirely based on localization data and previously reported effects of phospho-alleles, while more direct means to follow the phosphorylation state of the respective proteins (e.g. shift on an SDS- or Phos-tag gel etc,) and how they are influenced by other alleles are not used. The microscopy and the genetic combination of alleles may simply reach limits here in terms of what can be learned mechanistically. A more satisfying explanation for the relationship between Dis1 and Klp9 would require additional biochemical methods. Some questions that don't appear fully answered are: Is Clp1 really responsible for dephosphorylation of Dis1? Is it really the phosphorylation state of Klp9 that is influenced by the Dis1-6A mutation?Minimally, this section should be re-written to make it more accessible for the reader and more clearly explain the proposed mechanism. Ideally, a few key points should be tested experimentally: e.g. Phosphorylation state of Klp9 in different mutants by western blotting.

Due to the observation that *clp1* deletion results in the same localization pattern of Dis1-EGFP, as the expression of phospho-mimetic Dis1^6E^-EGFP, we strongly believe that Clp1 dephosphorylates Dis1. Moreover, the result that the reduced recruitment of Klp9-mCherry upon expression of phospho-deficient Dis1^6A^-EGFP can be rescued by simultaneous expression of phospho-deficient Klp9^3A^ suggests that Dis1 does regulate the phosphorylation state of Klp9. Besides, we do agree that further biochemistry analysis would be useful. However, since the time frame of Klp9 or Dis1 being present in their dephosphorylated form is very short (anaphase B takes approx. 15 min) and regarding the fact that these two proteins are phosphorylated on multiple other sites, it is very difficult to detect a change in the phosphorylation state by classical biochemistry methods. Therefore, we could not perform these experiments. However, we do agree that this section was rather confusing. We rewrote the section and tried to make it more clear.

4. Related to the previous point, it was considered a pity that the authors went away from the "half-spindle" assay in the later part of the paper (after Figure 3). Maybe some of the regulatory aspects would become more clear in this setting: Is Clp1 required for polymerization of the half spindle? What about the recruitment of Klp9 to the tip of the bundle? Is that impaired in Dis1-6A, is it increased in the Klp9-3A mutant, etc..? The authors could use this simplified setting more. It seems that the endpoint of the mechanism, presumably involving a dephosphorylation cascade that originates at Dis1, is really Klp9. The observation that Klp9-3A can override the Dis1-6A phenotype would support this notion. Therefore a closer investigation of Klp9-3A or -3E in this setting could be informative.

We fully agree with the reviewers. We have performed the experiments in the *cut7-24* mutant and added it to the section. We have also generated a phospho-mimetic Klp9^3D^ version. Unfortunately, the intensity of Klp9^3D^mCherry in the nucleus is much higher than in wild-type cells or cells expressing Klp9^3A^mCherry (see Author response image 1). Potentially, this mutant version of Klp9 cannot be degraded, leading to an increased abundance of the kinesin-6. This does not allow us to use the phospho-mimetic Klp9^3D^-mCherry for further analysis.

**Author response image 1. sa2fig1:** 

5. Results, Figure 6. Compared to the experiments performed in cells, the in vitro work appears somewhat preliminary.(a) Please show a Coomassie gel of purified Klp9.

The Gel was previously included in the supplementary figures: Figure 6—figure supplement 1.

Additional experiments may be worth considering to answer the following 2 points:(b) Is Klp9 processive, does it accumulate at microtubule ends? Is its motor activity required for its effect on microtubule dynamics? Please use fluorescently labelled Klp9 to visualize it's localization/motility?(c) Is Klp9 soluble at the higher tubulin concentration? Could it be that the effect of slowing down microtubule growth could be due protein insolubility (which could easily be seen when fluorescently labelled Klp9 is used)

For (b): We have tried to set up the experiments with Klp9-EGFP. Unfortunately, we did not succeed so far. In the tested construct the EGFP tag appears to alter motor function. Thus, we will not be able to solve these problems within the required time frame for the revisions of this paper.

However, to overcome the concern that the observed effect may not originate from Klp9’s motor activity as well as the concern mentioned in point (c) we performed the polymerization assays in presence of the non-hydrolysable ATP analog AMP-PNP. In these assays we could not observe an effect on the microtubule growth velocity upon Klp9 addition in the condition of high or low tubulin concentration, suggesting that indeed the motor activity of Klp9 is required for the regulation of microtubule growth. The additional results were included in the according section.

The following issue should be addressed in the Discussion:(d) How relevant is the observed behavior at 20 μm tubulin where the microtubule growth speed is significantly faster as in cells? In cells Klp9 clearly only accelerates microtubule growth speed according to the data presented in this study.

We have included our hypothesis in the discussion.